

# Extreme blocking ridges are associated with large wildfires in England

Kerryn Little[1], Dante Castellanos-Acuna[2], Nicholas Kettridge[1], Mike Flannigan[2], Piyush Jain[3]

[1]School of Geography, Earth and Environmental Sciences, University of Birmingham, Birmingham, United Kingdom
[2] Department of Natural Resource Science, Thompson Rivers University, Kamloops, British Columbia, Canada
[3] Northern Forestry Centre, Canadian Forest Service, Natural Resources Canada, Alberta, Canada

*Correspondence to:* Kerryn Little (k.e.little@bham.ac.uk)

**Abstract.** Persistent positive anomalies in 500 hPa geopotential heights (PPAs) are an event-based paradigm for tracking specific large scale atmospheric patterns that often correspond to blocking events. PPAs are associated with hot, dry surface weather conditions that promote fuel aridity and wildfire activity. We examine the importance of PPA events for surface fire weather across the UK and wildfires in England, a temperate, emerging fire prone region. Surface fire weather is more extreme under PPAs, characterised by reduced precipitation and anomalously high temperatures. Overall, 34% of England's burned area and 16% of all wildfire events occur during or up to five days following the presence of a PPA event. PPAs are generally more strongly associated with wildfire burned area than ignition frequency. The percentage of PPAs associated with wildfire events increases with increasing fire size, with PPAs being associated with half of wildfire events > 500 ha. PPAs are most important for heathland/moorland (40% burned area) followed by grassland (30% burned area) wildfires and are more important during the summer wildfire season. Synoptic-scale indicators of wildfire activity like PPAs may improve longer-term fire weather forecasts beyond surface fire weather indices alone, aiding wildfire preparedness and management decision-making. This is particularly important in emerging fire prone regions where wildfire risk is increasing but established tools for assessing fire danger may not yet exist.

**Short summary.** We demonstrate the importance of Persistent Positive Anomalies in 500 hPa Geopotential Heights (PPAs) for fire weather and wildfires in a temperate, emerging fire prone region using comprehensive wildfire occurrence records. PPAs become increasingly important for larger wildfires and are most important for heathland/moorland and grassland wildfires. Our findings demonstrate the potential of synoptic indicators for extending forecasting tools to aid wildfire preparedness and management.

## 1 Introduction

### 1.1 Wildfire risk in emerging fire prone regions

Wildfire risk is increasing in temperate, mid-latitude regions, exacerbated by land use and climate changes (Ellis *et al.* 2022; Jones *et al.* 2022). Wildfires are typically fuel limited in these regions, such as in the United Kingdom, due to the mild, humid climate that means fuels are often too wet to burn (Belcher *et al.* 2021). However, we are

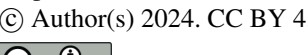



seeing increased fuel availability due to heatwaves and drought events as well as changes in land management
practices that may favour fuel aridity and accumulation, promoting wildfire activity (Glaves *et al.* 2020). Wildfires
tend to be smaller in temperate 'emerging fire prone' regions compared to more historically fire prone regions, as
highly fragmented landscapes and a high population density interfacing landscapes create fuel heterogeneity and
fast suppression response times. However, wildfires do not need to be large to be impactful (Belcher *et al.* 2021;
Kirkland *et al.* 2023; Stoof *et al.* 2024). Temperate peatlands contain globally important carbon stores that are
vulnerable to smouldering combustion and carbon emissions during severe wildfires (Page and Baird 2016;
Kirkland *et al.* 2023). Significant wildfires in the United Kingdom in recent years have resulted in ecological
impacts, carbon emissions, impacts to human health, and threat to homes and people as many people live within
the rural–urban interface (Davies *et al.* 2013; Glaves *et al.* 2020; Belcher *et al.* 2021; Naszarkowski *et al.* 2024).
Wildfire occurrence research in emerging fire prone regions has typically been limited because studies mainly
rely on satellite records that omit the majority of wildfires and detailed historical records often do not exist
(Fernandez-Anez *et al.* 2021). Existing understanding of fire–weather relationships has largely been developed
within more traditionally fire prone regions that have a long history of experiencing large and extreme wildfires
(e.g., Canada, Southern Europe, Australia, USA etc) and then adapted and applied within emerging fire prone
regions, usually to enable predictions of surface fire weather (e.g., de Jong *et al.* 2016; Masinda *et al.* 2022;
Steinfeld *et al.* 2022).

**1.2 Synoptic controls on surface fire weather**
In the midlatitudes, surface weather is broadly driven by synoptic-scale weather patterns (i.e., large scale upper-
air atmospheric circulation patterns (Franzke *et al.* 2020)). While surface weather is highly spatiotemporally
variable and difficult to forecast beyond the short-term, synoptic-scale upper-air (500 hPa) atmospheric patterns
can be more reliably predicted in the medium range (+10 days) (Hohenegger and Schär 2007). As such,
considering synoptic-scale indicators of wildfire activity in addition to surface fire weather may provide additional
insights for improving near-to-medium range forecasting of wildfire danger to aid wildfire preparedness and
management decision-making.

Geopotential height anomalies at the 500 hPa level help identify high-pressure blocking and ridge patterns that
are near stationary. As their name suggests, atmospheric blocking patterns block usual zonal airflow, and their
persistence can lead to dry, clear-sky conditions and high surface temperatures that may be amplified by land–
atmosphere feedbacks (Rex 1950). Such conditions promote fuel aridity and consequently wildfire activity
(Sharma *et al.* 2022). Persistent atmospheric blocking events can lead to synchronous elevated wildfire danger
across large areas, which can overwhelm wildfire response capabilities (Abatzoglou *et al.* 2021; Jain *et al.* 2024).

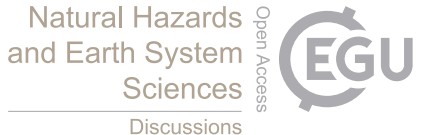

### 1.3 Synoptic controls on wildfire

Within Europe, previous research examining large-scale weather patterns associated with wildfire activity have tended to focus on countries in the Mediterranean (e.g., Duane and Brotons 2018; Pineda *et al.* 2022; Rodrigues *et al.* 2022), likely due to the history of significant wildfires and comprehensive wildfire occurrence databases; though there are some exceptions that examine Europe-wide (Giannaros and Papavasileiou 2023; Little, Castellanos-Acuna, *et al.* 2024) and Northern European (Wastl *et al.* 2013; Drobyshev *et al.* 2021) relationships. While atmospheric blocking has been associated with wildfire activity across Southern Europe, many studies have also highlighted the importance of strong wind events and atmospheric instability as drivers of extreme wildfire activity (Ruffault *et al.* 2017; Resco de Dios *et al.* 2022; Artés *et al.* 2022). Far fewer studies have examined large-scale atmospheric drivers of fire weather and wildfire activity within historically non-fire prone countries like the United Kingdom, where wildfire risk is increasing but established tools to predict fire weather and wildfire occurrence are often lacking. In these regions, extended periods of atmospheric blocking are likely important to sufficiently dry out fuels for wildfires.

Persistent Positive Anomalies in 500 hPa geopotential heights (PPAs) are an event-based paradigm for tracking extremes in high pressure blocking patterns (exceeding a threshold strength, size, and duration) through space and time. PPAs have recently been associated with extreme fire weather (hot, dry weather as defined by the Canadian Fire Weather Index) and wildfires in the Northern Hemisphere mid-latitudes for both Western North America (Sharma *et al.* 2022; Jain *et al.* 2024) and Europe (Little, Castellanos-Acuna, *et al.* 2024). More broadly, PPAs have also been associated with heatwaves and drought events across Europe (e.g., Robine *et al.* 2008; Pfahl and Wernli 2012; Tuel *et al.* 2022; Rousi *et al.* 2022).

We recently established the importance of PPAs for wildfires at a pan-European scale using the EFFIS burned area database (Little, Castellanos-Acuna, *et al.* 2024). We found that wildfires were more than twice as likely to occur during PPA events across Europe and were associated with 53% of burned area for Western Europe. However, the EFFIS burned area product only includes wildfires of around 30 ha and greater detected by satellite imagery, which resulted in very few records in regions with predominantly small wildfires. Notably, for the period March–October 2010–2020, EFFIS reported 348 wildfires for the UK (San-Miguel-Ayanz *et al.* 2012). In comparison, for the same period, the Fire and Rescue Service incident database reported 291,963 wildfires occurring in England alone (Forestry Commission 2023). There is a need to fully understand the importance of PPAs for wildfires in temperate regions like the UK where fires are predominantly smaller than those detectable by satellite products but can nonetheless be impactful. Using a comprehensive database of wildfire occurrence also allows us to examine seasonality and land cover dependent relationships between PPAs and wildfires, beyond the constraints of larger wildfires that may occur preferentially during summer on specific land covers.

We examine the importance of PPAs for fire weather and wildfire by addressing the following research questions: (1) what is the association between PPAs and surface fire weather across the UK between March–October 2001–2021? We then narrow in on the importance of PPAs for wildfires using a shorter but more comprehensive wildfire



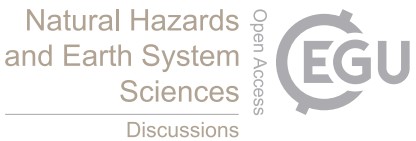

database for England: (2) what is the association between PPAs and wildfire activity for key land cover types in
England between March–October 2010–2020?

## 2 Methods

### 2.1 Study region

In this study, we first examine PPA associations with fire weather for the whole of the UK from 2001–2021. As
wildfire data were only available for England (Scotland, Wales, and Northern Ireland maintain their own incident
recording systems that may differ in data collection protocols (Fernandez-Anez *et al.* 2021)) for full years from
2010–2020, we examine PPA associations with wildfire for England only from 2010–2020 (hence the temporal
and regional difference in PPA–fire weather and PPA-wildfire analyses in this study).
Wildfires are a semi-natural hazard in the UK as ignitions are almost entirely anthropogenic (Gazzard *et al.* 2016).
Human use of fire on the landscape has been a traditional practice for centuries in the UK, particularly as a tool
for land management and habitat creation, and fire can bring positive ecological benefits (Belcher *et al.* 2021).
However, the risk of severe wildfires is increasing alongside changes in land use and climate (Belcher *et al.* 2021;
Arnell *et al.* 2021; Perry *et al.* 2022).

On average, over 30,000 wildfires are recorded in England annually, the majority of which are less than 1 ha, but
episodic larger wildfires also occur (nearly 13,000 fires > 1 ha between 2010–2020). The number of recorded
wildfires is highest within built-up areas and gardens, followed by arable, grassland, and woodland land covers.
However, the majority of burned area in England occurs in heathland/moorlands and grasslands (Forestry
Commission 2023). England experiences two main fire seasons, one in spring when shrub fuel moisture is lowest
following winter dormancy and prior to green-up, and a secondary season in mid-to-late summer (Fig. 1; Belcher
*et al.* 2021).

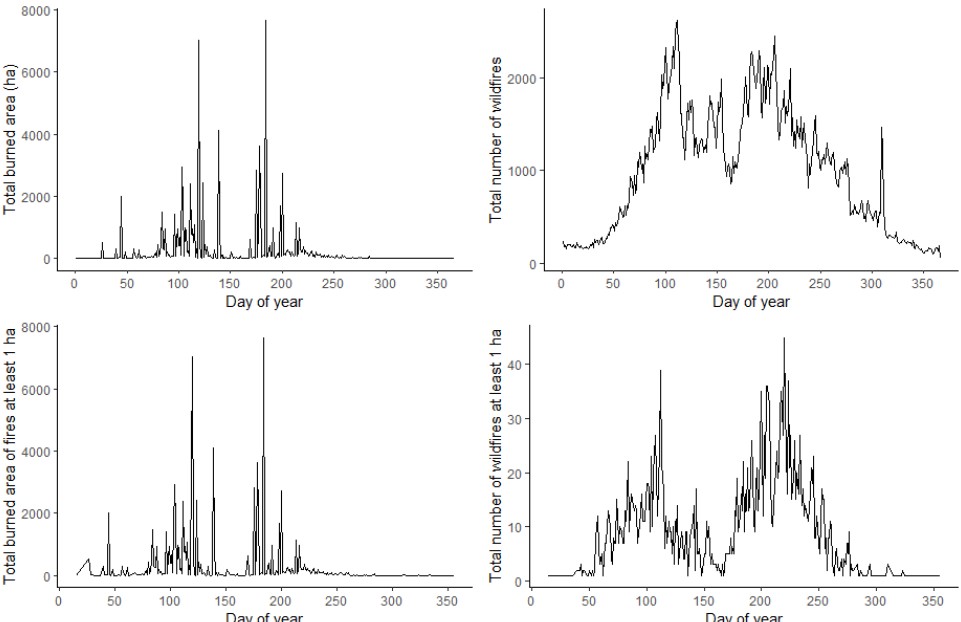

**Figure 1. Total daily (a) burned area, (b) number of wildfires, (c) burned area of wildfires at least 1 ha in size, and (d) number of wildfires at least 1 ha in size recorded in England between 2010–2020.**





**2.2 Data**
**2.2.1 Atmospheric data**
To identify PPA events, we obtained gridded 500 hPa geopotential height (Z500) from the European Centre for
Medium-Range Weather Forecasts (ERA5) global reanalysis dataset for the spatial domain 30°N to 75°N and -
50°E to 60°E from March–October 2001–2021 (Hersbach *et al.* 2020). We also obtained hourly surface
accumulated precipitation, 2-m air temperature and relative humidity, and 10-m U and V wind components (from
which we calculated wind speed) from ERA5 for the same period. We calculated the six components of the
Canadian Fire Weather Index System (CFWIS) using midday values of air temperature, wind speed, relative
humidity, and 24-h accumulated precipitation. The CFWIS describes the effects of surface weather on fuel
moisture and potential fire behaviour (Van Wagner 1987). The fine fuel moisture code (FFMC), duff moisture
code (DMC), and drought code (DC) are the fuel moisture indices of the CFWIS, which describe the moisture
conditions of the fine litter layer on the forest floor, top layer of organic material, and deeper soil layer,
respectively. These codes have increasing equilibrium nominal times of 16-h for the FFMC, 15 days for the DMC,
and 52 days for the DC representing the different drying rates of these fuels. The moisture codes include the
previous day's value as an input, thereby incorporating antecedent conditions into the CFWIS (Flannigan *et al.*
2016). The FFMC, DMC, and DC then feed into the fire behaviour components of the CFWIS. The initial spread
index (ISI) describes the potential rate of fire spread from the combined influence of wind speed and the FFMC,
and the build-up index (BUI) combines the DMC and DC to describe the amount of fuel available to burn. The
final fire weather index (FWI) combines the ISI and BUI to provide a measure of the overall potential fire intensity



(Van Wagner 1987). We used hourly values of air temperature and relative humidity to calculate the vapour
pressure deficit (VPD), which describes the ability of the atmosphere to extract moisture from dead fuels. We
calculated anomalies of all variables by subtracting daily values in each grid cell from the long-term climatological
mean (2001–2021). We also re-gridded all variables from 0.25x0.25 to 1x1 degree spatially and aggregated hourly
values to the daily mean, which is an appropriate resolution for identifying synoptic patterns (Barnes *et al.* 2012;
Liu *et al.* 2018; Sharma *et al.* 2022).

**2.2.2 PPAs**
There are various methods of identifying atmospheric blocking patterns, such as measuring the reversal of
meridional flow (Tibaldi and Molteni 1990; Pinheiro *et al.* 2019), dynamic potential vorticity (Pelly and Hoskins
2003; Small *et al.* 2014), or persistent positive anomalies in geopotential heights (Dole and Gordon 1983; Miller
*et al.* 2020). Because atmospheric blocking can occur without strong reversal of meridional flow and we are
interested in capturing events during the main wildfire season when strong polar dynamics are not common, we
opted to use the latter approach, which is less constrained by the specific blocking mechanism. This approach
allows us to capture the potential persistent, weaker pressure gradient events that characterise hot, dry surface
conditions for wildfires in summer (Sousa *et al.* 2018; Woollings *et al.* 2018). Furthermore, the definition of
persistent anomalies allows us to capture the most extreme instances of atmospheric blocking that strongly impact
surface conditions for wildfire from a background of weaker events (Woollings *et al.* 2018; Sharma *et al.* 2022;
Jain *et al.* 2024).

We calculated persistent positive anomalies (PPAs) of 500 hPa geopotential height using the detection algorithm
of Sharma *et al.* (2022). We identified PPA events for the pan-European spatial domain (-50°E to 60°E, 30°N to
75°N) as in (Little, Castellanos-Acuna, *et al.* 2024) and then filtered the database to include only PPA events with
grid-cells overlapping the UK between March–October 2001–2021. A full description of the PPA algorithm can
be found in Sharma *et al.* (2022) and (Little, Castellanos-Acuna, *et al.* 2024). Briefly, we calculated daily Z500
anomalies for each grid cell, applying a 5-day moving mean and weighted anomalies by the sine of latitude to
account for atmospheric energy dispersion (Dole and Gordon 1983). We used the daily varying mean standard
deviation of the Z500 anomaly in a 4-week moving window to calculate a seasonally varying threshold for
magnitude that allows us to capture the weaker pressure gradients that occur during the main wildfire seasons
compared to winter. We identified grid cell Z500 anomalies that exceeded a magnitude of 1.5x the daily varying
mean standard deviation of the Z500 anomaly for a duration of at least five days. We identified PPA events by
tracking the geometric centroid of spatially contiguous PPA grid cells until they reached a minimum size of 40,000
km$^2$. For each grid cell, days in which a PPA event directly overlaps the grid cell are identified as PPA days
(versus all other days, which are non-PPA days). We assessed the sensitivity of the algorithm to different Z500
anomaly magnitude thresholds (1, 1.5, and 2 standard deviations) to assess trade-offs between the strength of the
anomaly and its persistence in relation to wildfire activity (Table S1).



### 2.2.3 Wildfire data

We used the database of wildfire incidents attended by the Fire and Rescue Services in England for the period March–October 2010–2020 provided by the Home Office and quality checked by the Forestry Commission (Forestry Commission 2023). The Fire and Rescue Services Wildfire Operational Guidance defines a wildfire incident as 'any uncontrolled vegetation fire that requires a decision or action regarding suppression' (Scottish Government 2013). Information on each incident attended by the Fire and Rescue Service is recorded in this database, of which we extracted approximate location, start and end date of incident, landcover type burned, and burned area. Burned area of wildfires 1 ha and larger were validated by the Forestry Commission (Forestry Commission 2023), though we do not impose a minimum size threshold on the main analysis to capture the potential significance of small wildfires in the UK.

We aggregated all individual incident data to calculate total burned area and number of wildfires daily within each 1x1 grid cell. The main analyses use these daily burned area and number of wildfires per grid cell data, and 'fire days' were identified for each grid cell daily where burned area is recorded. We also repeated the above steps but first applying a minimum fire size threshold of greater than 1 ha; 10 ha; 50 ha; 100 ha; and 500 ha and filtering by wildfires on specific key UK landcovers (heathland/moorland; grassland (grassland, pasture, grazing); conifer forest; broadleaf forest; and standing crops) before summarising to the grid cell level to examine PPA–fire relationships across different wildfire sizes and landcover types.

We categorised each day as a PPA–fire, PPA–nofire, noPPA–fire, or noPPA–nofire combination for each 1x1 grid cell across England. We defined PPA–fire days as the presence of a PPA in a grid cell during or up to five days prior to that grid cell burning. This lag is to account for the role of PPA conditions in pre-drying fuels that subsequently ignite (supported by Figure S1 and previous research (Sharma *et al.* 2022)). (Little, Castellanos-Acuna, *et al.* 2024) assessed the sensitivity of PPA–fire associations across Europe to a range of different time lags, finding no major differences in the results. It should be noted that incident burned area is assigned to the incident start date as there is no daily breakdown of burned area within individual events. We acknowledge this limitation; however, as 99.9% of all wildfires and 72.7% of burned area are from incidents that occur within a single day or up to five days length, we believe this metric still largely captures whether incidents are associated with PPA events (particularly as PPA–fire days are defined by a five-day lag period).

### 2.3 Statistical analyses

We calculated PPA strength as the daily summed area-weighted magnitude of the 500 hPa anomaly. We then calculated the lead-lag relationship between PPA strength and daily surface anomalies across the UK. For each PPA, we calculated spatially averaged surface anomalies for the area of the PPA at maximum strength for 15 days prior to and following maximum PPA strength. The composite of all PPA events shows the response of surface conditions to the evolution of PPA events.

We developed linear regression models for every grid cell and for each month across the UK to assess whether there are differences in surface weather anomalies (dependent continuous variable) when a PPA is present over


the grid cell compared to when there is no PPA present. The monthly mean t-statistic from the linear model for
each grid cell tells us whether anomalies are larger (positive values) or smaller (negative values) on PPA days
than non-PPA days and the p-value identifies if the differences are statistically significant. Using the t-statistic of
the slope to present the results allows us to compare values across different variables.

Finally, we examined the association between PPA events and wildfires in England, by calculating the percentage
of burned area and number of wildfires that occurred during or up to five days following the presence of a PPA
in the grid cell compared to the total burned area and number of wildfires recorded annually, monthly, by specific
landcover types, and for different minimum thresholds of wildfire size. All statistical analyses were completed in
R version 4.1.2 (R Core Team 2022) using the packages igraph (Csardi and Nepusz 2006), Raster (Hijmans 2022a),
Terra (Hijmans 2022b), and zyp (Bronaugh and Werner 2019).
**3 Results**
**3.1 PPAs across the UK**
We detected 141 PPA events occurring over the UK landmass between March and October 2001–2021, averaging
6.7 events per year (Fig. 2). The average duration of a PPA event was 13.9 days (Table S1). PPA grid cells are
predominantly centred over (e.g., June), to the west (e.g., August), or the northeast (e.g., July) of the UK. PPAs
are least frequent in May.

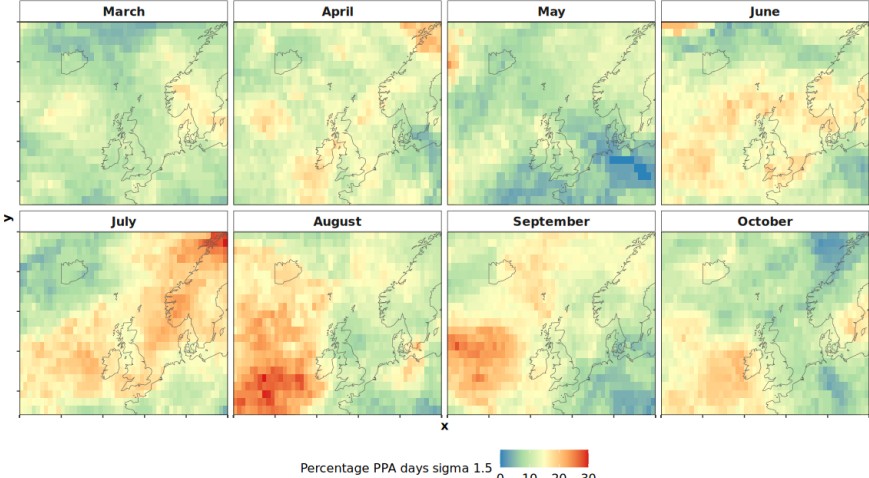

**Figure 2: Average monthly percentage PPA days (%) for each 1x1 degree grid cell in the spatial domain -30°E to 30°E**
**and 47°N to 70°N surrounding the United Kingdom between 2001–2021.**

**3.2 Surface fire weather is more extreme during PPA events**
Daily temperatures increase with increasing PPA strength and decline following peak PPA strength (Fig. 3). Wind
speed anomalies are lowest coincident with peak PPA strength, and precipitation anomalies are lowest in the four


days following peak PPA strength. However, relative humidity anomalies do not follow this pattern, and VPD is
driven by temperature rather than changes in atmospheric moisture during the PPA. The CFWIS components
increase with increasing PPA strength; however, there is a lag after maximum PPA strength before anomalies
peak and then a slower decline in the days following due to the slower response times of the fuel moisture codes
(equilibrium drying times of 16-h for FFMC; 15 days for DMC; and 52 days for DC).

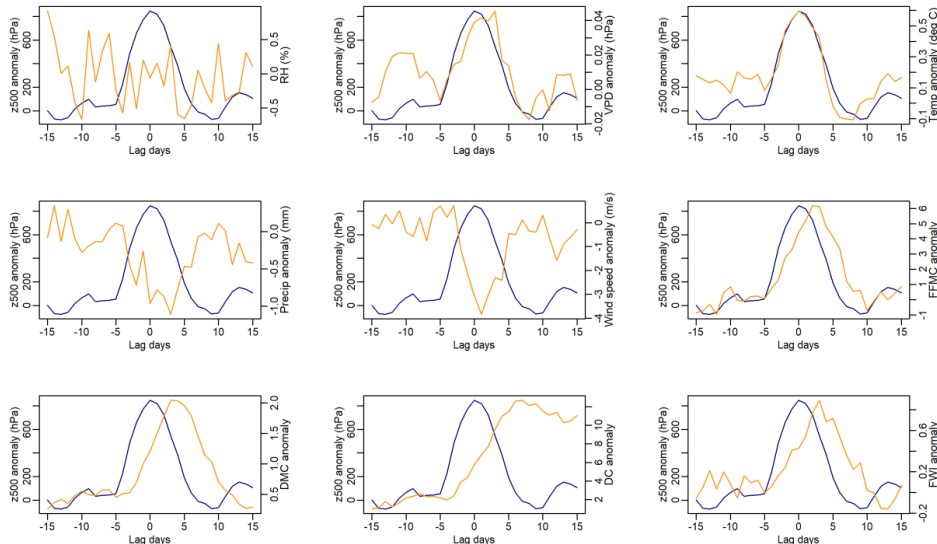

**Figure 3: Lead-lag relationship between PPA strength and surface anomalies. Blue line = z500 geopotential height**
**anomaly for PPA strength with maximum strength on day 0. Orange line = average surface anomaly for the maximum**
**PPA strength area 15 days either side of maximum PPA strength for (top left to bottom right) relative humidity, vapour**
**pressure deficit, temperature, precipitation, wind speed, FFMC, DMC, DC, and FWI anomalies.**
We present differences in surface weather anomalies between PPA and non-PPA days for the month of June here
(Fig. 4), one of the UK's peak summer wildfire months, with the results for all months presented in the
Supplementary Material (Fig. S2–S4) for brevity. Surface air temperature and VPD anomalies tend to be
significantly higher across the UK in June when there is a PPA present, while precipitation and wind speed
anomalies tend to be lower (Fig. 4). Components of the CFWIS are higher when there is a PPA present in June,
except for DC anomalies, which are higher across Scotland and Northern Ireland but lower across England and
Wales during PPA conditions. Differences in surface anomalies between PPA and non-PPA days are greatest in
June and July, while spring surface weather is not anomalous under PPA conditions, including precipitation
anomalies and the CFWIS fuel moisture codes, which are neither higher nor lower during PPAs in April (Fig. S2–
S4).



**Figure 4: (a)** t-statistic for linear regression models comparing the Canadian fire weather index (FWI) and fuel moisture codes (FFMC, DMC, DC), vapour pressure deficit (VPD), and temperature (Temp) anomalies during PPA days compared to non-PPA days for each 1x1 grid cell over the UK in June. All months and variables are shown in Fig. S2–S3. t-statistics > 0 (orange) show grid cells where surface anomalies are higher when there is a PPA present (larger t-statistics indicate a larger difference between PPA and non-PPA days). t-statistics < 0 (purple) show grid cells where surface anomalies are lower on PPA days. Significant differences ($P < 0.05$) are marked by a dot in the corresponding grid cell. **(b)** Boxplots showing the range of t-statistics for grid cell linear regression models of surface variable anomalies between PPA and non-PPA days across the UK in June (For each boxplot, the centre line is the median, the box is the interquartile range, and the upper and lower limits are maximum and minimum values, respectively). t-statistics > 0 indicate larger positive anomalies when there is a PPA present. Boxplots for all months shown in Fig. S4.



### 3.3 PPAs and wildfire activity across England

Overall, 34% of England's burned area occurs under and within five days of PPA conditions. There is significant monthly (Fig. 5) variability in the importance of PPAs for burned area. PPAs are most important for burned area in June (95%) followed by July (56%). The percentage of burned area associated with PPA events increases when a minimum fire size threshold is applied from 1 ha (35%), 10 ha (37%), 50 ha (39%), 100 ha (40%) to 500 ha (44%) (Table 1). The association between PPAs and the number of wildfires recorded is much lower overall at 16%, but again increases when a minimum fire size threshold is applied up to 500 ha (48%) (Table 2). When a weaker Z500 anomaly magnitude threshold of 1 x SD is used to define PPAs, the percentage of burned area associated with PPAs increases dramatically in April up to 79% but remains similarly high in June and July (Table S2).

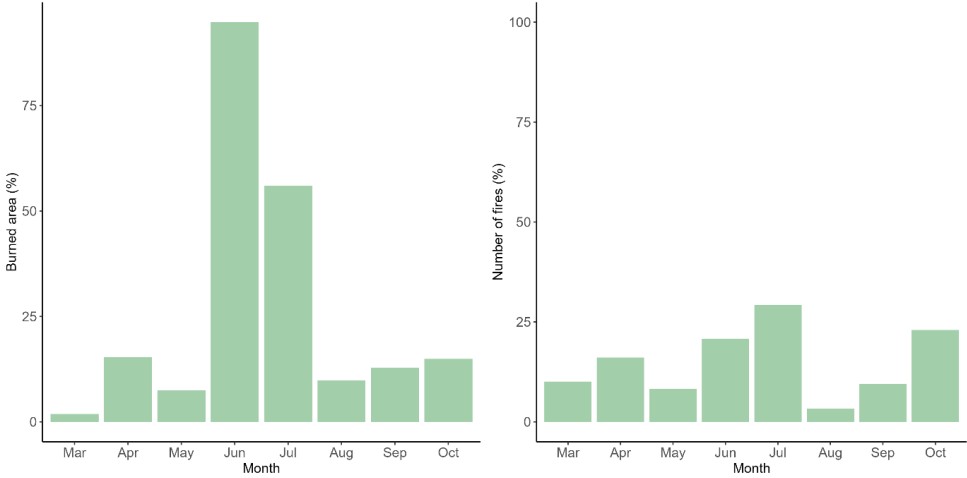

**Figure 5: Monthly percentage of burned area (left) and number of fires (right) in England associated with PPA events between March–October 2010–2020.**





**Table 1: Percentage burned area associated with PPA events when different minimum area burned thresholds are used**
**from March–October and annually. Total area burned (ha) associated with PPA events are given in brackets.**

| Month | All fires | > 1 ha fires | > 10 ha fires | > 50 ha fires | > 100 ha fires | > 500 ha fires |
|---|---|---|---|---|---|---|
| Mar | 2% (109 ha) | 1% (76 ha) | 1% (54 ha) | 0% (0 ha) | 0% (0 ha) | 0% (0 ha) |
| Apr | 15% (3307 ha) | 15% (3198 ha) | 15% (3092 ha) | 15% (2948 ha) | 15% (2648 ha) | 15% (2118 ha) |
| May | 7% (734 ha) | 7% (701 ha) | 7% (680 ha) | 7% (650 ha) | 8% (650 ha) | 9% (650 ha) |
| Jun | 95% (9902 ha) | 97% (9811 ha) | 98% (9737 ha) | 99% (9630 ha) | 100% (9430 ha) | 100% (8930 ha) |
| Jul | 56% (9947 ha) | 57% (9691 ha) | 59% (9368 ha) | 61% (8893 ha) | 63% (8840 ha) | 50% (8390 ha) |
| Aug | 10% (509 ha) | 10% (453 ha) | 11% (372 ha) | 14% (200 ha) | 0% (0 ha) | 0% (0 ha) |
| Sep | 13% (128 ha) | 13% (94 ha) | 24% (88 ha) | 100% (88 ha) | No fires | No fires |
| Oct | 15% (31 ha) | 2% (3 ha) | 0% (0 ha) | 0% (0 ha) | No fires | No fires |
| Annual | 34% (24667 ha) | 35% (24027 ha) | 37% (23391 ha) | 39% (22409 ha) | 40% (21568 ha) | 44% (20088 ha) |


**Table 2: Percentage number of fires associated with PPA events when different minimum area burned thresholds are**
**used from March–October and annually. Total number of fires associated with PPA events are given in brackets.**

| Month | All fires | > 1 ha fires | > 10 ha fires | > 50 ha fires | > 100 ha fires | > 500 ha fires |
|---|---|---|---|---|---|---|
| Mar | 10% (2896) | 5% (9) | 4% (2) | 0% (0) | 0% (0) | 0% (0) |
| Apr | 16% (7836) | 16% (39) | 22% (14) | 23% (8) | 23% (5) | 33% (3) |
| May | 8% (3390) | 9% (10) | 6% (2) | 8% (1) | 11% (1) | 25% (1) |
| Jun | 21% (7185) | 39% (30) | 65% (13) | 90% (9) | 100% (7) | 100% (6) |
| Jul | 29% (16730) | 31% (109) | 35% (24) | 38% (6) | 63% (5) | 50% (3) |
| Aug | 3% (1311) | 10% (37) | 10% (9) | 22% (2) | 0% (0) | 0% (0) |
| Sep | 10% (2529) | 4% (4) | 8% (1) | 100% (1) | No fires | No fires |
| Oct | 23% (3618) | 7% (1) | 0% (0) | 0% (0) | No fires | No fires |
| Annual | 16% (45495) | 16.3% (239) | 19% (65) | 25% (27) | 31% (18) | 48% (13) |


### 3.4 PPAs and wildfire burned area across key landcover types

Over 40% of all area burned from wildfires on heathlands / moorlands occurs during PPA events, including nearly
all burned area in June (98%) (Fig. 6). The association with wildfire ignitions is much lower, with 17% of wildfires
on heathland / moorlands occurring during PPAs. A total of 30% of burned area on grassland (including grassland,
pasture, and grazing) occurs during PPAs, while the association between burned area and PPA conditions is lowest
for broadleaf forest (19%), standing crops (15%), and conifer forest (7%). While percentage burned area during
PPAs shows substantial variability between landcover types and month-to-month, PPAs are associated with a
similar percentage of wildfire ignitions across all landcovers, ranging from 13% (standing crops) to 18%
(grassland).



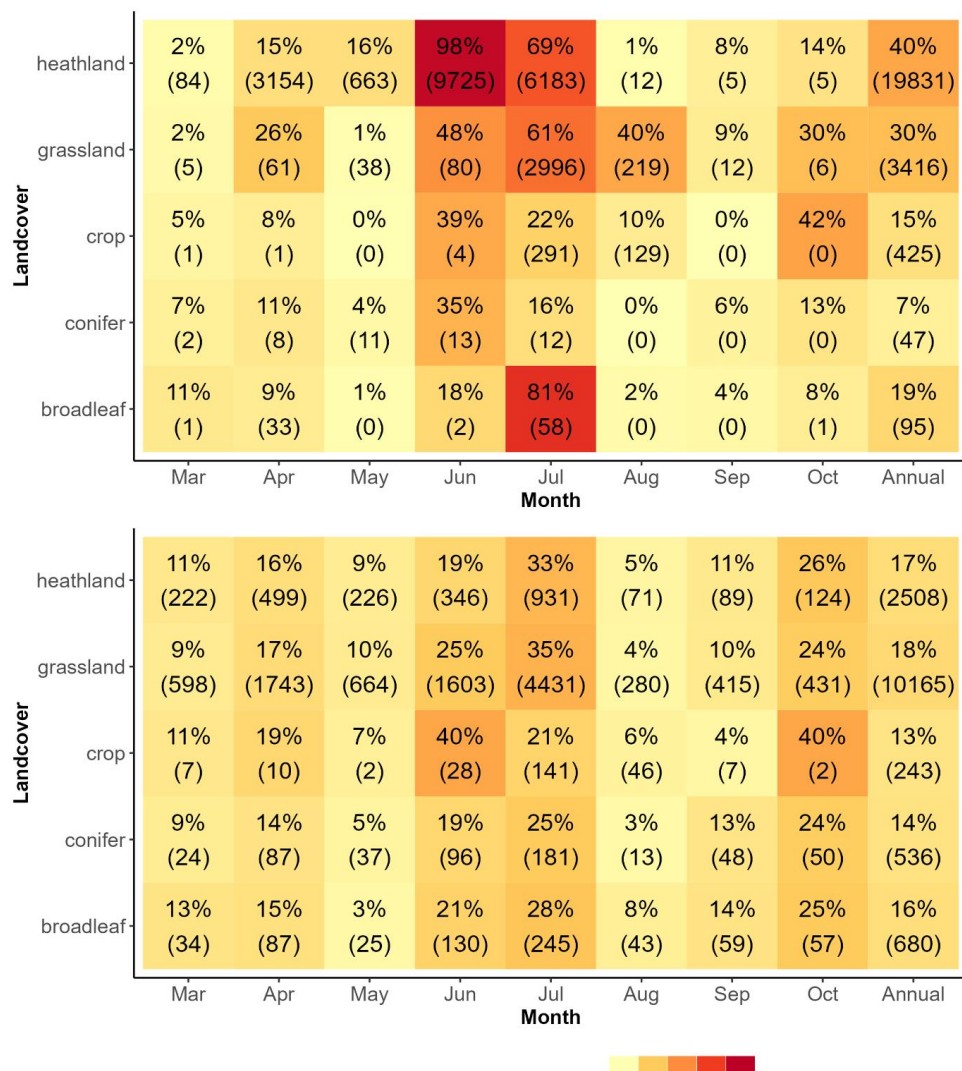

**Figure 6: Heat maps showing monthly percentage of burned area (top panel) and number of fires (bottom panel) associated with PPA events for key UK landcover types. Tile text describes the percentage followed by the total amount of burned area associated with PPAs in hectares (top panel) and total number of fires associated with PPAs (bottom panel) in brackets.**

## 4 Discussion

### 4.1 Large-scale drivers of fire weather in the UK and wildfires in England

We have demonstrated the importance of PPA events for elevated surface fire weather in the UK and as a driver

of wildfires in England. Our results agree with other studies that have linked atmospheric blocking to heatwaves


and droughts (McCarthy *et al.* 2019; Kay *et al.* 2020) and wildfires in the UK (Sibley 2019), as well as PPA events
specifically for North America and Europe (Sharma *et al.* 2022; Little, Castellanos-Acuna, *et al.* 2024; Jain *et al.*
2024). But our findings provide additional insights into when different types of fuels burn during PPA events
towards understanding the drivers of wildfire occurrence in England. Burned area and number of wildfires are
greatest in the zero to five days following PPA presence, which is consistent with the lag found with PPA events
Europe-wide (Little, Castellanos-Acuna, *et al.* 2024). However, compared to other parts of Europe and North
America, relative humidity anomalies are much smaller (Sharma *et al.* 2022; Jain *et al.* 2024). This is possibly
due to the moderating influence of a maritime climate where high atmospheric moisture prevents the development
of extremes that continental climates experience.

The relationship between PPAs and wildfire occurrence and burned area is strongest for large wildfire events. It
is acknowledged that there are few recorded large wildfire events in England to date, so large burned areas > 500
ha are predominantly driven by few events, notably the Saddleworth and Winter Hill fires of 2018 (Sibley 2019),
and these results should be interpreted cautiously. However, the importance of PPAs for annual number of fires
and burned area increases across all wildfire size thresholds applied (from 1 ha through to 500 ha), and as the
threat from wildfires increases in the UK, we may see more of these large wildfires in the future (Arnell *et al.*

366 2021).


**4.2 Seasonal role of PPAs for wildfire on different landcovers**
England experiences two main wildfire seasons, one in spring and one in summer, associated with different drivers
and fuels (Belcher *et al.* 2021; Nikonovas *et al.* 2024). While PPAs are important for wildfire burned area in June
(95% of burned area is associated with PPAs) and July (56%), they are less important in April (15%). In April,
wildfire occurrence is high on heathlands and moorlands. In these environments, live fuel moisture can often be
very low following winter dormancy and prior to new growth during the 'green-up' period and these fuels can
form an important part of the fuel load for wildfire spread (Davies *et al.* 2010). Weather conditions associated with
PPAs appear to be less important for heathland/moorland burned area during this spring season when phenological
controls dominate (Nikonovas *et al.* 2024). However, when a weaker threshold of geopotential height anomaly
(1x SD) is used to define PPAs, such as was used in Sharma *et al.* (2022), the importance of PPAs for wildfire
burned area in April increases to 79%. This would suggest that while extreme geopotential height anomalies are
less important in spring, persistent atmospheric blocking may still be an important circulation pattern for spring
wildfires in England.

During summer, weather conditions are more conducive to wildfire activity in general (high temperatures and dry
conditions can be experienced without requiring a PPA); however, for certain landcovers where live fuels are
important for wildfire spread, e.g., heathlands or moorlands, live fuel moisture content is often too high for
ignitions to occur (Little, Graham, *et al.* 2024). In these situations, we hypothesise that PPA events create extended
periods of elevated fire weather conditions that are needed to sufficiently dry fuels out for ignition. Furthermore,
extreme drying of fuels during PPA events may increase the continuity of available fuel for wildfire spread and
larger wildfires. While PPAs are important for area burned during wildfires, they are much less important for the



frequency of ignitions. This may be due to a combination of the complexity of different human-caused ignitions,
including non-climate factors such as geographic accessibility and socioeconomic factors, and the role of PPAs
in increasing continuity of available dry fuels for fires to burn larger areas but not necessarily increasing the
number of ignitions occurring.

The importance of PPA events for summer heathland and moorland fires in England suggests that strong wind
events may be less important for large wildfires in these fuels compared to other countries in Europe (Ruffault *et*
*al.* 2017; Carmo *et al.* 2022; Rodrigues *et al.* 2022), and it is the availability of fuel that is more of a control. This
builds on the work of (Little, Castellanos-Acuna, *et al.* 2024), which found that 95[th] percentile FWI anomalies
were more likely to occur during PPAs for Northern Europe compared to Southern Europe, as extreme FWI values
are often driven by wind speed (Dowdy *et al.* 2010).

To date, the dual spring and summer fire seasons have been a major challenge for predicting fire weather in the
UK as the fuels burning and their drivers differ seasonally (Belcher *et al.* 2021). Surface fire weather indices like
the FWI perform reasonably well in summer but fail to capture the spring fire season, while phenological
indicators improve spring fire season predictions but do not capture the heatwave events during summer that can
lead to extreme drying of fuels (Nikonovas *et al.* 2024; Ivison *et al.* 2024). It is important to consider all available
tools to accurately predict different periods of fire danger, including surface variables like weather, landcover,
and phenology, but also synoptic indicators.

**4.3 Forecasting and management implications**
Inclusion of synoptic indicators of wildfire activity like PPAs in wildfire occurrence prediction models provides
opportunities to extend forecasting capabilities, and previous studies have demonstrated skill in forecasting
geopotential heights (Weyn *et al.* 2019; He *et al.* 2019). Increasing global temperatures will lead to more extreme
geopotential height anomalies (He *et al.* 2024), which will increase the likelihood of large wildfires occurring
during these persistent hot, dry weather conditions (though how dynamic patterns will change in response to this
warming is still uncertain). PPAs provide an approach to detect and track these extremes in space and time in
order to forecast periods of elevated wildfire danger in the near-to-medium range. Jain *et al.* (2024) recently
demonstrated the role of PPAs during the 2021 heat dome event in western North America, finding this event
accounted for 21–34% of total burned area in 2021 and was partially attributed to climate change. A number of
studies have also demonstrated the ability of synoptic fire weather patterns to predict fire weather (e.g., Lagerquist
*et al.* 2017; Papavasileiou and Giannaros 2023; Humphrey *et al.* 2024). Synoptic fire weather forecasting may
provide insights for wildfire management decision-making, including resource allocation, to prevent suppression
resources from becoming overwhelmed, such as happened in the 2022 July wildfires in London when the UK's
first large-scale destruction of properties due to wildfires was experienced (London Fire Brigade 2023; John and
Rein 2024).

The times of year and land covers where PPAs are not important for wildfires are also insightful. PPAs detect
only the most extreme instances of atmospheric blocking (depending on the threshold anomaly used in the


detection algorithm), and other atmospheric circulation patterns may also be important. Furthermore,
understanding the drivers of days where a high number of ignitions but overall small burned area can overwhelm
response resources is also important. Future work will thoroughly examine the synoptic climatology of wildfires
in England to understand the synoptic indicators of wildfire beyond the hot, dry extremes of PPA events. This
will allow for the development of fire weather classes for medium-range forecasting applications, as has been
conducted for other countries (e.g., Skinner *et al.* 2002; Lagerquist *et al.* 2017; Rodrigues 2019; Zhong *et al.* 2020).
The UK does not have a fire danger rating system although the Met Office Fire Severity Index (MOFSI), an
adaptation of the Canadian FWI, is used to trigger land closures in England and Wales during the most extreme
conditions (Met Office 2023). Synoptic indicators of elevated wildfire danger periods would therefore provide
significant benefits for near-to-medium range wildfire preparedness and awareness in emerging fire-prone regions
like the UK where tailored assessments of fire danger do not yet exist.

**4.4 Conclusions**
Surface fire weather is more extreme across the UK and wildfires are larger in England during PPA events. Our
results provide insights into the seasonality and fuel type dependencies of PPA–wildfire relationships, finding:
(1) PPAs are strongly associated with wildfire burned area but not necessarily ignition frequency.
(2) PPA–wildfire relationships strengthen with increasing wildfire size.
(3) Strong wind speeds and atmospheric moisture anomalies associated with wildfires in continental fire
prone regions may be less important for wildfires in maritime regions like the UK.
(4) PPAs are most important for wildfire burned area in summer and for heathland/moorland and grassland
wildfires.
**Code and data availability**
We used the R algorithm of Sharma *et al.* (2022) to identify PPAs over the UK landmass, and we direct the reader
to this reference for further information. All climate datasets used in this paper are publicly available through the
references  cited  and  can  be  accessed  through  the  web  portals
https://cds.climate.copernicus.eu/cdsapp#!/dataset/reanalysis-era5-complete?tab=overview  (ERA5  reanalysis
data  for  surface  and  upper  air  variables,  accessed  January  2022)  and
https://zenodo.org/record/626193#.X9pTFdhKg4s (CFWIS data, accessed January 2022). Wildfire records from
the Home Office Incident Recording System for England can be obtained by contacting the Home Office UK.
**Author contributions**
All authors were involved in conceptualisation and writing – review & editing. KL, DCA, and PJ designed the
methodology, curated the data, and validated the analyses. KL conducted the formal analyses and wrote the
original draft.



**Competing interests**
The authors declare that they have no conflict of interest.
**Acknowledgements**
The authors would like to thank Aseem Sharma for providing the R script of the PPA algorithm used in Sharma
*et al.* (2022) that we used to identify PPAs. This project has received funding from the NERC Highlight project
NE/T003553/1.

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
