# Peer review of "Extreme blocking ridges are associated with large wildfires in England"

_Natural Hazards and Earth System Sciences, 2024_

## Author Comment (AC1)

**Reviewer 1**

*Thank you for engaging with our manuscript and for your constructive feedback. We have responded to your comments below in blue with \*, with substantial changes proposed to the manuscript text being indicated in red with "".

Little et al. explore the relationship between persistent high-pressure systems, surface weather conditions, fire indicators and actual wildfires using a dataset which ranges from March – October 2001-2021 (UK) and 2010-2020 (England), respectively. They find significant relationships between high pressure systems and local weather conditions and wildfire characteristics across seasons. This is an interesting assessment, which could be further improved by accounting for following comments:

**Minor:**

1. Fig. 1 this is a very informative figure. It could be improved even further, by adding information on months and seasons by e.g. dashed vertical lines or shaded background.

*Thank you for the suggestion, we have now updated Fig. 1 to be more informative and have moved it to become Fig. 4 in the results:

[Figure]

"Figure 4: Total log transformed daily (a) burned area and (b) number of vegetation fires recorded in England between 2010–2020. Day of year on the x-axis is partitioned to show the calendar months (dashed lines) winter (DJF), spring (MAM), summer (JJA) and autumn (SON)."

2. L. 144 is this the best way to implement this? Wouldn't daily max. Temperatures be more informative? Or is this essentially the same value for most of the days.

*The standard implementation for calculating the Canadian Fire Weather Index System (CFWIS) components is to use noon values of surface variables (Van Wagner, 1987), this is what is used operationally (e.g., Vitolo et al. (2019)) and in existing datasets of the CFWIS available online

(e.g., McElhinny et al. (2020)) so we used the same for comparability. Historically the reason for using noon values is an operational consideration, seeing as many weather stations reported values at noon local standard time. The CFWIS consequently uses the noon values to calibrate fire danger at the peak burning period.

3.   L. 245 do these numbers refer to each grid-cell individually or are events defined based on some sort of spatial integration (all anomalous neighboring grid-cells are counted as one event)?

*They refer to events overall, which as you say are based on spatial integration of neighbouring cells. We define this in L187, which we have elaborated on slightly for clarity:

"We identified PPA events by tracking the geometric centroid of spatially contiguous PPA grid cells until they reached a minimum size of 40,000 km$^2$ after which we labelled it a PPA event."

4.   This does not refer to the content, but rather to the way the paper is structured. Data and Methods sections seem to include some results already, which is why the results section is fairly short (mostly the figure, the figure caption and 1-2 sentences of description). The manuscript would be easier to follow if methods and related results were merged.

*We have now removed the study region section of the methods, moving the relevant introductory context to a new section of the introduction that describes the fire situation in the UK (1.2 Vegetation fires in the UK) and moving the figure and associated summary of wildfire statistics to a new section of the results (3.3 Vegetation Fires in England).

"1.2 Vegetation fires in the UK

Vegetation fires are a semi-natural hazard in the UK as ignitions are almost entirely anthropogenic (Gazzard *et al.* 2016). Human use of fire on the landscape has been a traditional practice for centuries in the UK, particularly as a tool for land management and habitat creation, and fire can bring positive ecological benefits (Belcher et al., 2021); however, there is evidence that the risk of severe vegetation fires is increasing (Arnell et al., 2021; Belcher et al., 2021; Perry et al., 2022). In the UK, Fire and Rescue Services Wildfire Operational Guidance defines a vegetation fire incident as 'any uncontrolled vegetation fire that requires a decision or action regarding suppression' (Scottish Government, 2013). Currently, this does not impose a minimum size threshold on the definition of a vegetation fire, and indeed wildfires do not need to be large to be impactful (Belcher et al., 2021; Kirkland et al., 2023; Stoof et al., 2024). The UK has a high population density of 280 people per square km, compared to traditionally fire prone countries like Australia (3 people per square km), Canada (4 people per square km) and USA (36 people per square km) (World Bank, 2022). This means that natural landscapes are highly fragmented and lack the fuel continuity to generate massive burned areas, and fires tend to be detected quickly. Moreover, a high population density means that a high proportion of fires occur in the interface between people, infrastructure and environment. Vegetation fires in these areas can threaten lives and property, despite their often small size (Graham et al., 2020; John and Rein, 2024; London Fire Brigade, 2023). Critically, vegetation fire preparedness still lags behind other countries and response capabilities can be overwhelmed in extreme conditions (John and Rein, 2024; Climate Change Position Statement, 2025; Pandey et al., 2023)."

"3.3 Vegetation fires in England

On average, over 30,000 vegetation fires are recorded in England annually, the majority of which are less than 1 ha, but episodic larger fires also occur (nearly 13,000 fires > 1 ha between 2010–2020). The number of recorded fires is highest within built-up areas and gardens, followed by arable, grassland, and woodland land covers. However, the majority of burned area in England occurs in heathland/moorlands and grasslands (Forestry Commission, 2023). England experiences two main fire seasons, one in spring

when shrub fuel moisture is lowest following winter dormancy and prior to green-up, and a secondary season in mid-to-late summer (Fig. 4; Fig. S5; Belcher *et al.* 2021).

[Figure]

Figure 4: Total log transformed daily (a) burned area and (b) number of fire occurrences recorded in England between 2010–2020. Day of year on the x-axis is partitioned to show the calendar months (dashed lines) winter (DJF), spring (MAM), summer (JJA) and autumn (SON)."

5.    Figure 3. It would be helpful (also for the description of the results in the paragraph before and for the caption) to have the subplot labeled with e.g. letters a-i. Abbreviations should be written out fully in the caption.

*Thank you for this suggestion, we have updated Figure 3 to be clearer (note, this is now Figure 2 in the revised manuscript).

[Figure]

"Figure 2: Lead-lag relationship between PPA strength and surface anomalies. Blue line = z500 geopotential height anomaly for PPA strength with maximum strength on day 0. Orange line = average surface anomaly for the maximum PPA strength area 15 days either side of maximum PPA strength for (top left to bottom right) (a) noon relative humidity, (b) maximum vapour pressure deficit, (c) noon temperature, (d) 24-h accumulated precipitation, (e) noon wind speed, and (f) FFMC, (g) DMC, (h) DC, and (i) FWI anomalies."

6.    Figure 4. The subplot labeling and the figure caption could be made more intuitive by labeling each plot separately. Abbreviated variables should be written out. A clear description of what the boxplot boundaries, whiskers and dots refer to is missing.

*We have also updated Figure 4 to address this lack of clarity (note, this is now Figure 3 in the revised manuscript).

[Figure]

"Figure 3: (a–f) t-statistic for linear regression models comparing anomalies of the Canadian fuel moisture codes (a) drought code (DC), (b) duff moisture code (DMC), and (c) fine fuel moisture code (FFMC) (d) the overall Canadian fire weather index (FWI); (e) daily noon temperature; and (f) daily max vapour pressure deficit (VPD) during PPA days compared to non-PPA days for each 1x1 grid cell over the UK in June. All months and variables are shown in Fig. S2–S3. t-statistics > 0 (orange) show grid cells where surface anomalies are higher when there is a PPA present (larger t-statistics indicate a larger difference between PPA and non-PPA days). t-statistics < 0 (purple) show grid cells where surface anomalies are lower on PPA days. Significant differences (*P < 0.05*) are marked by a dot in the corresponding grid cell. (g) Boxplots showing the range of t-statistics for grid cell linear regression models of surface variable anomalies between PPA and non-PPA days across the UK in

June (For each boxplot, the centre line is the median, the box is the interquartile range, dots are outliers, and the upper and lower whiskers are maximum and minimum values, respectively). From left to right, the variables presented are the Canadian Fire Weather Index System build-up index (BUI), drought code (DC), duff moisture code (DMC), fine fuel moisture code (FFMC), fire weather index (FWI), and initial spread index (ISI); and daily total precipitation (PREC), daily noon relative humidity (RH); daily noon temperature (TEMP); daily maximum vapour pressure deficit (VPD); and daily noon wind speed (WIND). t-statistics > 0 indicate larger positive anomalies when there is a PPA present. Boxplots for all months shown in Fig. S4."

7. I would suggest to merge the conclusions into the discussion section and possibly add a bit more context.

*After considering your comment, we have opted to remove the conclusion section (which wasn't really adding anything) in favour of rounding out the last section of the discussion on implications to come back to PPAs:

"The UK does not have a fire danger rating system although the Met Office Fire Severity Index (MOFSI), an adaptation of the Canadian FWI, is used to trigger land closures in England and Wales during the most extreme conditions (England and Wales Fire Severity Index, 2023). The evolution of PPA events through 500 hPa geopotential height forecasts could be tracked to identify periods of sustained elevated fire weather that may challenge or overwhelm available resources for fire management (Jain et al., 2024). Synoptic indicators of elevated wildfire danger periods would therefore provide significant benefits for near-to-medium range wildfire preparedness and awareness in emerging fire-prone regions like the UK where tailored assessments of fire danger do not yet exist."

**Major:**

1. How are PPA events being associated with surface anomalies? Are relationships being quantified for events that are detected over the same grid-point, during the same time period only? How are the lead lag relationships accounted for and how are vertical shears in PPAs and surface response integrated in the regression shown in Fig. 4?

*We look at the association between PPA events and surface anomalies in multiple ways. Figure 2 in the revised manuscript looks exclusively at lagged variables, which presents a composite of spatially averaged surface anomalies for the area of the PPA at maximum strength for 15 days prior to and following maximum PPA strength. This shows the response of surface anomalies to the evolution of PPA events. We then conduct the linear regression analysis for differences in surface anomalies for each grid cell monthly on days there is a PPA present over the grid cell compared to days where there is no PPA present, for the same time period 2001–2021 (note this is now Fig. 3 in the revised manuscript). The inclusion of the FWIS indices accounts for lagged weather in this analysis through the time lags applied to the fuel moisture codes FFMC (16-h), DMC (15 days), and DC (52 days). Because PPAs are defined by a single level of 500 hPa geopotential heights, it is not meaningful to discuss vertical shears in PPAs.

2. The Dataset is fairly short for a trend analysis, but it would be very insightful to acknowledge and assess changes in wildfire characteristics over the past decade(s). Have increased temperatures, precipitation or landcover changes led to significant increases (or decreases in regional wildfires?). An increased threat from Wildfires to the UK is mentioned in l.363 but is not backed up with any quantitative results.

*As you state, the dataset is short for a trend analysis and a presentation of wildfire statistics for the UK has already been published in Forestry Commission (2023) "Wildfire statistics for England: Report to 2020-21" so we do not look to comprehensively cover this topic within this paper and instead reference previous studies to support these statements. However, a trend analysis would be very interesting to consider within future work looking at surface fire weather and wildfires in the UK, and we will look to take this comment onboard in future work.

3.  Various fire weather indicators were calculated in this study. Which indicator is best suited to predict actual fires in the UK?

*This is a very interesting question that is being considered in an ongoing study that focuses more on the actual vegetation fire dataset and surface fire weather in the UK. We agree that there is a need to clearly establish the relationship between surface fire weather and vegetation fires in this manuscript but wish to keep the focus of the manuscript on the core PPA analysis and prevent the manuscript from becoming overly long. We propose to include a section within the results presenting the distribution of fire weather indicators for vegetation fire occurrences by season and fire size, to show the following: (1) surface fire weather is anomalously high when vegetation fires occur, (2) surface fire weather is highest for larger fires, (3) there are differences in which indicators are higher during vegetation fires, namely the FFMC in spring compared to the DC and BUI in summer. The stronger performance of the FFMC in spring compared to the FWI in summer is supported by previous research, which we have added references to in the discussion section.

"3.4 Surface fire weather and vegetation fire occurrence across England

Surface fire weather, as indicated by the CFWIS indices, tends to be anomalously high when vegetation fires occur in England (Fig. 5). This is true for both spring and summer fires less than and greater than 50 ha, though fire weather anomalies are highest for larger fires in summer. In spring, the FFMC is the most elevated when vegetation fires occur, while the other indices are distributed around the average or slightly above average in the case of the FWI. In summer, the DC and BUI are anomalously high when vegetation fires occur and the FWI is positively skewed for large fires but not fires < 50 ha.

[Figure]

Figure 5: Density plots showing the distribution of the Canadian Fire Weather Index System indices anomalies: (a) FWI, (b) FFMC, (c) DMC, (d) DC, (e) ISI and (f) BUI for spring (left subpanel) and summer (right subpanel). Distributions are presented for fires less than or equal to 50 ha (blue fill shows the distribution, dashed blue vertical line shows the mean value) and fires greater than 50 ha (green fill shows the distribution, dashed green vertical line shows the mean value). The black vertical line at zero separates positive anomalies (indices are higher than average during fires) from negative anomalies (indices are below average during fires). Note the independent axes labels for readability."

Addition to the discussion in Section 4.2:

"To date, the dual spring and summer fire seasons have been a major challenge for predicting fire weather in the UK as the fuels burning and their drivers differ seasonally (Belcher et al., 2021). Surface fire weather indices like the FWI subcomponent of the CFWIS perform reasonably well in summer but fail to capture the spring fire season (Davies and Legg, 2016; de Jong et al., 2016; Nikonovas et al., 2024), while phenological indicators (for example, vegetation greenness indices like the Enhanced Vegetation Index 2 (EVI2) (Nikonovas et al., 2024)) improve spring fire season predictions but do not capture the heatwave events during summer that can lead to extreme drying of live fuels (Ivison et al., 2024; Nikonovas et al., 2024). It is important to consider all available tools to accurately predict different periods of fire danger, including surface variables like weather, landcover, and phenology, but also synoptic indicators."

---

## Author Comment (AC2)

**Reviewer 2**

*Thank you for taking the time to give feedback on our manuscript. We have responded to your comments below in blue with a *, with substantial proposed changes to the manuscript text being indicated in red with "". Given your concern regarding the methodology and terminology of wildfires, we would like to address this first here before moving to the specific comments. We do not believe there are flaws in the methodology or interpretation of results, and we hope that this explanation will resolve any concerns – A lack of clarity in the manuscript may have contributed to this misunderstanding and we thank you for the opportunity to address this in the revised manuscript.

Respectfully, we disagree that small fires and fires in built-up areas and gardens should be excluded from this manuscript, though we thank you for bringing to our attention that this should be better justified and communicated throughout our manuscript. In England, due to a high population density and landscape fragmentation, there are no truly 'wild' or 'natural' areas as one might expect to find elsewhere like North America (Gazzard et al., 2016). Fires will always be smaller because of this, but this is also exactly why small fires and fires surrounding urban areas are important (arguably potentially more so than larger fires in remote locations) because they can easily put people and property in danger. A notable large-scale destruction of homes from vegetation fire in the UK was 'just' 20 ha in size (Wennington Fire – 2022). At the same time, a single 20 ha fire in a crop field resulted in £40,000 damage to one farm. National Farmers Union Mutual (a UK insurance company) reported costs of £110 million from farm and vehicle fires in the UK in 2023, despite the average crop fire burning 1.5 ha (unpublished data).

The term 'wildfire' appears to be causing some confusion regarding our manuscript as there is no globally agreed definition and it spans across many different scales of fire occurrences. This is a very interesting discussion that would make a nice opinion piece but is well outside the scope of our own research. Instead, we have now opted to remove the use of the term wildfire in this manuscript, and instead we refer to 'vegetation fire occurrence'. We hope this provides complete clarity in the focus of this manuscript and avoids the potential for confusion introduced by preconceived perceptions of the term wildfire that the reader may have (e.g., Tedim and Leone, 2020). We have also clearly defined vegetation fire occurrences in the context of our manuscript using the UK Fire and Rescue Services Wildfire Operational Guidance definition: "A wildfire is any uncontrolled vegetation fire that requires a decision or action regarding suppression" (Scottish Government, 2013). We have moved this definition earlier in the manuscript and have elaborated on our discussion of the importance of small vegetation fires and the UK situation in the introduction section to clarify what it is we are interested in and why it is relevant very early on.

Similarly, we have not discriminated on the landcover on which the vegetation fire burned because fires within urban vegetated areas pose a significant threat to people and property. Sadly, the recent fires in California are testament to the importance of this now more than ever. Again, we have placed greater emphasis on why these fires are important in the introduction.

Rather than removing all small fires and specific landcovers, we instead explored these relationships within the manuscript, demonstrating how relationships with PPAs change when using different thresholds of vegetation fire size (using all fires, > 1 ha, > 10 ha, > 50 ha, > 100 ha, > 500 ha) annually and by month. We also look at the number of fires in each of these categories to be very clear about the sample size when considering different thresholds. Section 3.5 (previously section 3.4) also breaks down the relationships according to key landcover types,

which excludes fires in urban areas and focuses on heathland/moorland, grassland, broadleaf forest, standing crops, and conifer forest – again providing all details on the number of fires in each category.

Within this review, Reviewer 2 suggested fires smaller than 10 ha, 30 ha or even 50 ha should be excluded, while Reviewer 3 suggested thresholds of 100 ha or 500 ha, which illustrates that applying a size threshold would be an arbitrary decision that there is not a consensus on. We believe that our analysis, which presents the results across key landcovers, months, and for different threshold sizes provides a much more thorough examination of the relationship between PPAs and vegetation fires than narrowing the scope to a single arbitrary size or landcover category. This in turn allows the reader to understand the nuances between fire size and number of fires in the data set when drawing conclusions.

Thank you for entertaining this somewhat lengthy comment, but we hope that this clarifies the decisions made in the manuscript and resolves any concerns the reviewers previously had. We appreciate the chance to improve our manuscript to more clearly articulate these points to the readership of NHESS. We have added a section on UK vegetation fires in the introduction to outline the vegetation fire context, define vegetation fires and introduce the significance of small fires:

**"1 Introduction**

**1.1 Vegetation fire risk in emerging fire prone regions**

Vegetation fire risk is increasing in temperate regions like the UK, which have historically experienced few large fires, due to mild, humid climates that mean fuels are generally less flammable (Belcher et al., 2021). However, changes in land management practices combined with a warming climate are increasing the quantity of live biomass available to burn (Glaves et al., 2020; Belcher et al., 2021). Research in these so-called "emerging fire-prone" regions has been limited because wildfires tend to be dominated by smaller fires, many of which are not detected in satellite records nor included in historical records (Fernandez-Anez et al., 2021). Consequently, decision support systems based on fire weather–fire relationships have primarily been developed in fire-prone regions that have a long history of experiencing large and extreme wildfires (e.g., Canada, Southern Europe, Australia, USA) and then adapted for use in emerging fire-prone regions (e.g., de Jong *et al.* 2016; Masinda *et al.* 2022; Steinfeld *et al.* 2022). Despite and because of these challenges, there is an urgent need to understand and quantify wildfire risk to inform long-term wildfire preparedness in these regions (Pandey et al., 2023).

**1.2 Vegetation fires in the UK**

Vegetation fires are a semi-natural hazard in the UK as ignitions are almost entirely anthropogenic (Gazzard *et al.* 2016). Human use of fire on the landscape has been a traditional practice for centuries in the UK, particularly as a tool for land management and habitat creation, and fire can bring positive ecological benefits (Belcher et al., 2021); however, there is evidence that the risk of severe vegetation fires is increasing (Arnell et al., 2021; Belcher et al., 2021; Perry et al., 2022). In the UK, Fire and Rescue Services Wildfire Operational Guidance defines a vegetation fire incident as 'any uncontrolled vegetation fire that requires a decision or action regarding suppression' (Scottish Government, 2013). Currently, this does not impose a minimum size threshold on the definition of a vegetation fire, and indeed wildfires do not need to be large to be impactful (Belcher et al., 2021; Kirkland et al., 2023; Stoof et al., 2024). The UK has a high population density of 280 people per square km, compared to traditionally fire prone countries like Australia (3 people per square km), Canada (4 people per square km) and USA (36 people per square km) (World Bank, 2022). This means that natural landscapes are highly fragmented and lack the fuel continuity to generate massive burned areas, and fires tend to be detected quickly. Moreover, a high population density means that a high proportion of fires occur in the interface between people,

infrastructure and environment. Vegetation fires in these areas can threaten lives and property, despite their often small size (Graham et al., 2020; John and Rein, 2024; London Fire Brigade, 2023). Critically, vegetation fire preparedness still lags behind other countries and response capabilities can be overwhelmed in extreme conditions (John and Rein, 2024; Climate Change Position Statement, 2025; Pandey et al., 2023).

**1.5 Rationale**

Fewer studies have explored the large-scale atmospheric drivers of fire weather and vegetation fire occurrence in countries historically not prone to fires, such as the UK. Although fire risk is increasing in these regions, established tools for predicting fire weather and fire occurrence are often lacking. In the UK, extended periods of atmospheric blocking likely play a key role in sufficiently drying out vegetation and elevating fire risk. To address this gap, it is crucial to better understand the role of PPAs in vegetation fires, particularly in temperate regions like the UK, where fires are often smaller than those detectable by satellite but still impactful. A comprehensive fire occurrence database allows us to examine the seasonality and land cover-dependent relationships between PPAs and vegetation fires, extending beyond the larger fires typically observed in summer on specific land covers.   We investigate the importance of PPAs for fire weather and vegetation fire occurrence by addressing the following research questions: (1) what is the association between PPAs and surface fire weather across the UK between March–October 2001–2021? Then, using the comprehensive vegetation fire occurrence database for England, (2) What is the association between PPAs and vegetation fire occurrence across key land cover types in England from March–October 2010–2020?"

First of all, I would like to apologize for the long delay in providing the revision of the manuscript.

The authors investigate the occurrence of extreme blocking ridges associated with large wildfires in England. For that they use a method to identify atmospheric blocking patterns for the pan-European spatial domain. The method is based on persistent positive anomies of 500 hPa geopotential height.

Main issues:

- The data provided by the Home Office for England, covering the period between March and October 2010–2020, requires careful consideration. First, the analysis should be restricted to fires exceeding 10 hectares, or preferably, 50 hectares. Fires below these thresholds generally have a minimal or negligible impact. If such thresholds are applied, the majority of the analysis and conclusions will be based on a limited number of cases, which restricts the generalizability of the findings.

*As discussed above, we respectfully disagree that all fires less than 10 or 50 ha generally have a minimal or negligible impact. Please see the overall comment for evidence of impactful vegetation fires in the UK less than 50 ha in size, such as the Wennington 2022 fire that destroyed ~20 properties, as well as changes we have made to the revised manuscript to address this concern. Table 1 and 2 give a comprehensive summary of how PPA–fire relationships change when different size thresholds are considered, as well as how many cases are included in each category, thereby allowing the reader to make informed conclusions about the generalisability of findings. We believe this to provide a more nuanced analysis than simply applying an arbitrary size threshold, which we note differs in recommendation between the reviewers too.

- Introduction: The introduction lacks clarity in several places. For example, lines 39–41 and 44–46 are vague and read more like a report rather than scientific writing. These

sections should be rewritten to align with scientific standards, ensuring precision and focus.

*We have rewritten the introduction to be more focused, including these sentences specifically and added further references to support the information, which now form part of the focused paragraph on small vegetation fires:

"Introduction

1.1 Vegetation fire risk in emerging fire prone regions

Vegetation fire risk is increasing in temperate regions like the UK, which have historically experienced few large fires, due to mild, humid climates that mean fuels are generally less flammable (Belcher et al., 2021). However, changes in land management practices combined with a warming climate are increasing the quantity of live biomass available to burn (Glaves et al., 2020). Research in these so-called "emerging fire-prone" regions has been limited because wildfires tend to be dominated by smaller fires, many of which are not detected in satellite records nor included in historical records (Fernandez-Anez et al., 2021). Consequently, decision support systems based on fire weather–fire relationships have primarily been developed in fire-prone regions that have a long history of experiencing large and extreme wildfires (e.g., Canada, Southern Europe, Australia, USA) and then adapted for use in emerging fire-prone regions (e.g., de Jong *et al.* 2016; Masinda *et al.* 2022; Steinfeld *et al.* 2022). Despite and because of these challenges, there is an urgent need to understand and quantify wildfire risk to inform long-term wildfire preparedness in these regions (Pandey et al., 2023).

1.2 Vegetation fires in the UK

Vegetation fires are a semi-natural hazard in the UK as ignitions are almost entirely anthropogenic (Gazzard *et al.* 2016). Human use of fire on the landscape has been a traditional practice for centuries in the UK, particularly as a tool for land management and habitat creation, and fire can bring positive ecological benefits (Belcher et al., 2021); however, there is evidence that the risk of severe vegetation fires is increasing (Arnell et al., 2021; Belcher et al., 2021; Perry et al., 2022). In the UK, Fire and Rescue Services Wildfire Operational Guidance defines a vegetation fire incident as 'any uncontrolled vegetation fire that requires a decision or action regarding suppression' (Scottish Government, 2013). Currently, this does not impose a minimum size threshold on the definition of a vegetation fire, and indeed wildfires do not need to be large to be impactful (Belcher et al., 2021; Kirkland et al., 2023; Stoof et al., 2024). The UK has a high population density of 280 people per square km, compared to traditionally fire prone countries like Australia (3 people per square km), Canada (4 people per square km) and USA (36 people per square km) (World Bank, 2022). This means that natural landscapes are highly fragmented and lack the fuel continuity to generate massive burned areas, and fires tend to be detected quickly. Moreover, a high population density means that a high proportion of fires occur in the interface between people, infrastructure and environment. Vegetation fires in these areas can threaten lives and property, despite their often small size (Graham et al., 2020; John and Rein, 2024; London Fire Brigade, 2023). Critically, vegetation fire preparedness still lags behind other countries and response capabilities can be overwhelmed in extreme conditions (John and Rein, 2024; Climate Change Position Statement, 2025; Pandey et al., 2023).

1.3 Synoptic controls on surface fire weather and vegetation fire

In the midlatitudes, surface weather is driven by synoptic-scale weather patterns (i.e., large scale upper-air atmospheric circulation patterns (Franzke et al., 2020)). While surface weather is highly spatiotemporally variable and difficult to forecast beyond the short-term, synoptic-scale upper-air (500 hPa) atmospheric patterns can be more reliably predicted in the medium range (+10 days) (Hohenegger and Schär, 2007). As such, considering synoptic-scale indicators of vegetation fire occurrence in addition to surface fire weather may provide additional insights for improving near-to-medium range forecasting of

fire danger to aid fire preparedness and management decision-making (Humphrey et al., 2024; Jain et al., 2024; Papavasileiou and Giannaros, 2023).

Within Europe, previous research examining large-scale weather patterns associated with vegetation fire occurrence have tended to focus on countries in the Mediterranean (e.g., Duane and Brotons 2018; Pineda *et al.* 2022; Rodrigues *et al.* 2022), likely due to the history of significant vegetation fires and comprehensive fire occurrence databases; though there are some exceptions that examine Europe-wide (Giannaros and Papavasileiou, 2023; Little et al., 2024) and Northern European (Drobyshev et al., 2021; Wastl et al., 2013) relationships. While atmospheric blocking has been associated with vegetation fire occurrence across Southern Europe, other studies have also highlighted the importance of strong wind events and atmospheric instability as drivers of extreme vegetation fire activity (Artés et al., 2022; Resco de Dios et al., 2022; Ruffault et al., 2017).

**1.4 Persistent positive anomalies in geopotential heights (PPAs)**

Atmospheric blocking occurs when a high pressure system remains nearly stationary such that it effectively "blocks" the usual mid-latitude zonal airflow, leading to dry, clear-sky conditions and high surface temperatures that may be amplified by land–atmosphere feedbacks (Rex, 1950). Such conditions promote fuel aridity and consequently vegetation fire occurrence (Sharma et al., 2022). Moreover, persistent atmospheric blocking events can lead to synchronous elevated fire danger across large areas, which can overwhelm fire response capabilities (Abatzoglou et al., 2021; Jain et al., 2024).

Positive geopotential height anomalies at the 500 hPa level are widely used to identify high-pressure blocking events (Tibaldi and Molteni, 1990). One such method, Persistent Positive Anomalies in 500 hPa geopotential heights (PPAs), are an event-based paradigm for tracking extremes in high pressure blocking patterns (that exceed a threshold amplitude, size, and duration) through space and time (Dole and Gordon 1983). Compared with other methods of identifying atmospheric blocking patterns, such as measuring the reversal of meridional flow (Pinheiro et al., 2019; Tibaldi and Molteni, 1990) and dynamic potential vorticity (Pelly and Hoskins, 2003; Small et al., 2014), PPAs are less constrained by the specific blocking mechanism, which can be important for capturing events during the main fire season when strong polar dynamics are not common (Dole and Gordon, 1983; Miller et al., 2020). PPAs can be used to capture the potential persistent, weaker pressure gradient events that characterise hot, dry surface conditions for fires in summer (Sousa et al., 2018; Woollings et al., 2018).

PPAs have recently been associated with extreme fire weather (hot, dry weather as defined by the Canadian Fire Weather Index) and vegetation fires in the Northern Hemisphere mid-latitudes for both Western North America (Jain et al., 2024; Sharma et al., 2022) and Europe (Little et al., 2024). We recently established the importance of PPAs for vegetation fires at a pan-European scale finding that fires were more than twice as likely to occur during PPA events across Europe and were associated with 53% of burned area for Western Europe (Little et al., 2024). However, the EFFIS burned area product used in that study only includes vegetation fires of around 30 ha and greater detected by satellite imagery, which resulted in very few records in regions with predominantly small vegetation fires. Notably, for the period March–October 2010–2020, EFFIS reported 348 vegetation fires for the UK (San-Miguel-Ayanz et al., 2012). In comparison, for the same period, the Fire and Rescue Service incident database reported 291,963 vegetation fires occurring in England alone (Forestry Commission, 2023).

**1.5 Rationale**

Fewer studies have explored the large-scale atmospheric drivers of fire weather and vegetation fire occurrence in countries historically not prone to fires, such as the UK. Although fire risk is increasing in these regions, established tools for predicting fire weather and fire occurrence are often lacking. In the UK, extended periods of atmospheric blocking likely play a key role in sufficiently drying out vegetation and elevating fire risk. To address this gap, it is crucial to better understand the role of PPAs in vegetation fires, particularly in temperate regions like the UK, where fires are often smaller than those detectable by satellite but still impactful. A comprehensive fire occurrence database allows us to examine the seasonality and land cover-dependent relationships between PPAs and vegetation fires, extending

beyond the larger fires typically observed in summer on specific land covers. We investigate the importance of PPAs for fire weather and vegetation fire occurrence by addressing the following research questions: (1) what is the association between PPAs and surface fire weather across the UK between March–October 2001–2021? Then, using the comprehensive vegetation fire occurrence database for England, (2) What is the association between PPAs and vegetation fire occurrence across key land cover types in England from March–October 2010–2020?"

- Blocking Paragraph (Lines 65–70): The explanation of blocking is not presented correctly. The paragraph begins with a discussion of the 500 hPa level, followed by a description of what blocking is. This structure is unclear and suggests a lack of understanding of the concept. A more structured and detailed explanation of atmospheric blocking is required, ensuring the authors demonstrate a thorough grasp of the subject.

*We agree with the reviewer that this section was not presented in the most logical and clear order. We have now restructured this section:

"1.3 Synoptic controls on surface fire weather and vegetation fire

In the midlatitudes, surface weather is driven by synoptic-scale weather patterns (i.e., large scale upper-air atmospheric circulation patterns (Franzke et al., 2020)). While surface weather is highly spatiotemporally variable and difficult to forecast beyond the short-term, synoptic-scale upper-air (500 hPa) atmospheric patterns can be more reliably predicted in the medium range (+10 days) (Hohenegger and Schär, 2007). As such, considering synoptic-scale indicators of vegetation fire occurrence in addition to surface fire weather may provide additional insights for improving near-to-medium range forecasting of fire danger to aid fire preparedness and management decision-making (Humphrey et al., 2024; Jain et al., 2024; Papavasileiou and Giannaros, 2023).

Within Europe, previous research examining large-scale weather patterns associated with vegetation fire occurrence have tended to focus on countries in the Mediterranean (e.g., Duane and Brotons 2018; Pineda *et al.* 2022; Rodrigues *et al.* 2022), likely due to the history of significant vegetation fires and comprehensive fire occurrence databases; though there are some exceptions that examine Europe-wide (Giannaros and Papavasileiou, 2023; Little et al., 2024) and Northern European (Drobyshev et al., 2021; Wastl et al., 2013) relationships. While atmospheric blocking has been associated with vegetation fire occurrence across Southern Europe, other studies have also highlighted the importance of strong wind events and atmospheric instability as drivers of extreme vegetation fire activity (Artés et al., 2022; Resco de Dios et al., 2022; Ruffault et al., 2017).

1.4 Persistent positive anomalies in geopotential heights (PPAs)

Atmospheric blocking occurs when a high pressure system remains nearly stationary such that it effectively "blocks" the usual mid-latitude zonal airflow, leading to dry, clear-sky conditions and high surface temperatures that may be amplified by land–atmosphere feedbacks (Rex, 1950). Such conditions promote fuel aridity and consequently vegetation fire occurrence (Sharma et al., 2022). Moreover, persistent atmospheric blocking events can lead to synchronous elevated fire danger across large areas, which can overwhelm fire response capabilities (Abatzoglou et al., 2021; Jain et al., 2024).

Positive geopotential height anomalies at the 500 hPa level are widely used to identify high-pressure blocking events (Tibaldi and Molteni, 1990). One such method, Persistent Positive Anomalies in 500 hPa geopotential heights (PPAs), are an event-based paradigm for tracking extremes in high pressure blocking patterns (that exceed a threshold amplitude, size, and duration) through space and time (Dole and Gordon 1983). Compared with other methods of identifying atmospheric blocking patterns, such as measuring the reversal of meridional flow (Pinheiro et al., 2019; Tibaldi and Molteni, 1990) and dynamic potential vorticity (Pelly and Hoskins, 2003; Small et al., 2014), PPAs are less constrained by the specific blocking mechanism, which can be important for capturing events during the main fire season when strong polar dynamics are not common (Dole and Gordon, 1983; Miller et al., 2020). PPAs can be used to

capture the potential persistent, weaker pressure gradient events that characterise hot, dry surface conditions for fires in summer (Sousa et al., 2018; Woollings et al., 2018).

PPAs have recently been associated with extreme fire weather (hot, dry weather as defined by the Canadian Fire Weather Index) and vegetation fires in the Northern Hemisphere mid-latitudes for both Western North America (Jain et al., 2024; Sharma et al., 2022) and Europe (Little et al., 2024). We recently established the importance of PPAs for vegetation fires at a pan-European scale finding that fires were more than twice as likely to occur during PPA events across Europe and were associated with 53% of burned area for Western Europe (Little et al., 2024). However, the EFFIS burned area product used in that study only includes vegetation fires of around 30 ha and greater detected by satellite imagery, which resulted in very few records in regions with predominantly small vegetation fires. Notably, for the period March–October 2010–2020, EFFIS reported 348 vegetation fires for the UK (San-Miguel-Ayanz et al., 2012). In comparison, for the same period, the Fire and Rescue Service incident database reported 291,963 vegetation fires occurring in England alone (Forestry Commission, 2023)."

> Section 1.3. Antecedent conditions are crucial for understanding most fire behavior. These conditions, including prolonged dry periods, accumulated fuel loads, and seasonal trends, significantly influence fire behavior and severity. The section should emphasize the role of these pre-existing factors in shaping fire activity.

*We believe this has been sufficiently addressed in Section 1.3, but we have added some further text to line 144 (Section 2.1.1) to emphasise the role of the FWI system in accounting for antecedent weather conditions:

"The moisture codes include the previous day's value as an input, thereby incorporating antecedent conditions of the last 50+ days into the CFWIS (Flannigan et al., 2016)."

- L97 Does it really matters analyzing fires below 30ha or even 50ha?

*Yes, we believe it does matter in this context, and we have tried to communicate this more clearly now for an international audience. Ground-truthed vegetation fire records beyond satellite imagery are really important for countries that don't routinely experience massive fires, and this dataset has not been analysed through this lens before. We have dedicated more of the introduction to communicating the importance of small fires in countries experiencing increasing vegetation fire risk as well as understanding the current fire situation to inform decisions surrounding future fire risk.

- Why is this study important and why is different from Little et al 2024??

*The importance of the analysis of a comprehensive database of UK vegetation fire occurrence records that have not previously been analysed in the academic literature should not be dismissed. We understand that by international standards this is a relatively short record (although 10 years of data is still a significant effort) but it allows us to explore an understudied area of wildfire research through small fires that aren't captured by current satellite capabilities. We have added a sentence to the vegetation fire data section and the rationale in the introduction to highlight the importance of this dataset.

Little et al. (2024) examined the role of PPAs for wildfires detected by the EFFIS burned area product, but this excludes wildfires < 30 ha, i.e., the majority of vegetation fires the UK experiences. Furthermore, Little et al. 2024 did not examine the key landcover types burned, nor the surface fire weather associated with vegetation fires, which we have added into the proposed results. The examination of the role of PPAs for different types of landcover burned and for different fire sizes, focusing on an understudied area (rather than Europe wide, where

nuanced relationships that exist at a subregional level are masked and overlooked compared to the strong (but extensively researched) fire–climate relationships within Mediterranean Europe) distinguish this piece of research from Little et al. 2024.

- L126 The analysis raises the question of why wildfires in built-up areas and gardens are being included. Are these incidents truly wildfires? By definition, wildfires typically occur in natural or semi-natural landscapes, and including these cases may lead to misleading interpretations or dilute the focus of the study.

*Please see our initial comment for a thorough discussion of this point, but to summarise, it depends on the definition used (and there are many definitions of wildfire). We used the definition used in the UK, where this research was carried out, and this is the definition that Fire and Rescue Services use to classify vegetation fire incidents they attend. The recent California fires sadly demonstrate that built-up areas and gardens will still burn, and while we do not expect similar behaviour in the UK by any means, vegetation fires surrounding properties pose one of the largest threats to people and infrastructure here. In saying this, we present the results across a range of different size thresholds, those above 1 ha exclude the majority of fires in built-up areas, but there is a negligible change in the percentage of burned area and number of fires associated with PPAs across these thresholds, suggesting the results are robust to their inclusion or exclusion. Further, the breakdown of relationships by landcover allows the reader to consider just heathland, grassland, crop or forest fires.

We believe that some of the concerns regarding this manuscript are around the 'wildfire' terminology, which is detracting from the main focus of the research. To improve the clarity and focus of the manuscript and avoid entering into debate around terminology, we have opted to remove the term 'wildfire' from the manuscript and instead refer to 'vegetation fire occurrence'. This highlights both that we refer to outdoor fires where vegetation is burning and that it is the occurrence of a fire (rather than the subsequent behaviour of the fire) that we are interested in.

- Figure 1. The total burnt area of specific large wildfires dominates the results, overshadowing smaller events and potentially skewing the analysis. A more detailed breakdown or normalization of the data might help clarify the trends and improve the interpretability of the figure.

*We have altered Figure 1 to be more informative and log transformed the axes, this has now become Figure 4 in the results section:

[Figure]

"Figure 4: Total log transformed daily (a) burned area and (b) number of vegetation fires recorded in England between 2010–2020. Day of year on the x-axis is partitioned to show the calendar months (dashed lines) winter (DJF), spring (MAM), summer (JJA) and autumn (SON)."

- Re-grid: I don´t understand how did you re-grid the data? Specially for the wild fires. If there is one wild fire with 1ha within a certain 1x1 degree box then you count this grid point with a wildfire? Do you think that´s a fair thing to do? If this is the case, I don´t agree with the methodology since a single wildfire with a 1ha is not meaningful on a 1x1 degree box.

*The re-gridding is actually only related to the PPA and surface weather analysis – thank you for bringing this to our attention as this is not how it is communicated in the text. Originally, we had looked at an analysis using PPA-fire grid cells, but this was not included in the final manuscript, so we have now removed the section on this in the text. The gridded data were used to identify the presence of a PPA event in that grid cell, which is appropriate for looking at synoptic scale phenomena (Liu et al., 2018; Sharma et al., 2022). For each wildfire, we then labelled it as being associated with a PPA if the coordinates were within a PPA grid cell, during or up to 5 days following the presence of a PPA. The number of lag days since PPA presence were also recorded. Otherwise, the fire was labelled as a non-PPA event. The percentage of burned area and fires associated with PPAs is therefore analysed by fire record rather than by grid cell (the opposite to how it was communicated originally). Therefore, a single fire with a small burned area will only have a small weighting on the results, which look at overall burned area and total number of fires. We now explain this more clearly in the text in Section 2.1.3:

"We labelled each incident as a PPA–fire event if a PPA was present in the grid cell the incident occurred in either on the same day or up to five days preceding the incident, otherwise it was labelled a noPPA–fire event. This five day lag is to account for the role of PPA conditions in pre-drying fuels that subsequently ignite (supported by Fig. S1 and previous research (Sharma et al., 2022)). (Little et al., 2024) assessed the sensitivity of PPA–fire associations across Europe to a range of different time lags, finding no major differences in the results. It should be noted that incident burned area is assigned to the incident start date as there is no daily breakdown of burned area within individual events. We acknowledge this limitation; however, as 99.9% of all vegetation fires and 72.7% of burned area are from incidents that occur within a single day or up to five days length, we believe this metric still largely captures whether incidents are associated with

PPA events (particularly as PPA–fire events are defined by a five-day lag period). We also repeated the above steps but first applying a minimum fire size threshold of greater than 1 ha; 10 ha; 50 ha; 100 ha; and 500 ha and filtering by fires on specific key UK landcovers (heathland/moorland; grassland (grassland, pasture, grazing); conifer forest; broadleaf forest; and standing crops) to examine PPA–fire relationships across different vegetation fire sizes and landcover types."

- I have another major issue using the specific detection blocking method. Using this specific method, you only have a yes or no analysis. My suggestion is to use Weather Regimes (Grams et al., 2017) instead of your blocking method. This as many advantages: 1) The use of weather regimes gives more opportunity to study other meteorological patterns that can be important for wildfires which you cannot do it in the present form. 2) the authors mentioned several times, that this study would be important for helping in forecasting wildfires. It the PPAs operational forecast in use? Another advantage of using Weather regimes is that they are operational forecasted at the ECMWF (https://www.ecmwf.int/en/newsletter/165/meteorology/how-make-use-weather-regimes-extended-range-predictions-europe). The authors could consider combining the Fire Weather Index (FWI) forecasts from Copernicus with weather regime forecasts to enhance wildfire predictability in England. This integrated approach could improve the accuracy of forecasting wildfire risks by leveraging the strengths of both systems, providing valuable insights for prevention and mitigation strategies.

Grams, C., Beerli, R., Pfenninger, S. et al. Balancing Europe's wind-power output through spatial deployment informed by weather regimes. Nature Clim Change 7, 557–562 (2017). https://doi.org/10.1038/nclimate3338

*We agree that analysing Weather Regimes is an interesting and useful research question and one we are currently working on in collaboration with the UK Met Office on a different project, using UK-specific weather typing. However, we believe this is a completely separate research question to the focus of this manuscript. It is not simply whether a blocking pattern is present or not that dictates conditions for vegetation fires, and blocking patterns occur alongside a wide range of different weather conditions. The purpose of the PPA algorithm is to pull out the most extreme conditions from week-to-week blocking weather patterns. That is, those events that are especially large, long-lasting, and anomalous in terms of their geopotential heights that we may expect to lead to larger numbers of fires occurring or larger burned areas. This method is therefore beneficial for assessing the extent to which these extreme conditions are important for fires and whether they should be specifically monitored for periods of elevated fire risk – such relationships would potentially be diluted if we used weather regimes instead. While the PPA forecast is not operational currently, there is no reason it could not be tracked using readily-available data in real-time to forecast the development of these extreme events that may require specific actions and preparations above what would be undertaken for a typical blocking event. We highlight this in the discussion of our research, but as a research paper, it is not necessary to consider only operational forecasts or models (and in fact, operational products need to be preceded by such underlying research papers).

**Considering these significant major issues, I recommend the rejection of the manuscript in its current form. I also suspect that the use of wildfires only above 10ha or 50ha and excluding in built-up areas and gardens areas will give a very limited sample for analysis which will limited the findings of the study. The points outlined above highlight critical flaws in the analysis, interpretation, and presentation of results, which need substantial revision before the work can be reconsidered again for publication.**

---

## Author Comment (AC3)

**Reviewer 3**

This manuscript aims to demonstrate the role of Persistent Positive Anomalies in 500 hPa Geopotential Heights (PPAs) for fire weather danger and wildfires in England, a temperate and evolving fire-prone region using wildfire occurrence records between 2010 and 2020. The authors base their study on a short database of fires in England. I have several criticisms related to methodology (the threshold for burned area and number of fires) and interpretation of results (concept of wildfire, fuel load and moisture and ignitions) listed below that should be addressed before considering the paper for publication in Natural Hazards and Earth System Sciences journal.

*Thank you kindly for your feedback on this manuscript. We have already thoroughly addressed your criticism of the methodology in the comments to Reviewer 2, and we hope this resolves your concerns too as we do not believe there are any methodological flaws in the manuscript. There were places where the manuscript lacked clarity, and we are grateful for the opportunity to more thoroughly justify our methodology in the revised manuscript. We have responded to your comments below in blue with a *, with substantial proposed changes to the manuscript text being indicated in red with "". We have pasted the overall comment to Reviewer 2 here for your convenience:

Given your concern regarding the methodology and terminology of wildfires, we would like to address this first here before moving to the specific comments. We do not believe there are flaws in the methodology or interpretation of results, and we hope that this explanation will resolve any concerns – A lack of clarity in the manuscript may have contributed to this misunderstanding and we thank you for the opportunity to address this in the revised manuscript.

Respectfully, we disagree that small fires and fires in built-up areas and gardens should be excluded from this manuscript, though we thank you for bringing to our attention that this should be better justified and communicated throughout our manuscript. In England, due to a high population density and landscape fragmentation, there are no truly 'wild' or 'natural' areas as one might expect to find elsewhere like North America (Gazzard et al., 2016). Fires will always be smaller because of this, but this is also exactly why small fires and fires surrounding urban areas are important (arguably potentially more so than larger fires in remote locations) because they can easily put people and property in danger. A notable large-scale destruction of homes from vegetation fire in the UK was 'just' 20 ha in size (Wennington Fire – 2022). At the same time, a single 20 ha fire in a crop field resulted in £40,000 damage to one farm. National Farmers Union Mutual (a UK insurance company) reported costs of £110 million from farm and vehicle fires in the UK in 2023, despite the average crop fire burning 1.5 ha (unpublished data).

The term 'wildfire' appears to be causing some confusion as there is no globally agreed definition and it spans across many different scales of fire occurrences. This is a very interesting discussion that would make a nice opinion piece, but this is well outside the scope of our own research. Instead, we have now opted to remove the use of the term wildfire in this manuscript, and instead we refer to 'vegetation fire occurrence'. We hope this provides complete clarity in the focus of this manuscript and avoids the potential for confusion introduced by preconceived perceptions of the term wildfire that the reader may have (e.g., Tedim and Leone, 2020). We have also clearly defined vegetation fire occurrences in the context of our manuscript using the UK Fire and Rescue Services Wildfire Operational Guidance definition: "A wildfire is any uncontrolled vegetation fire that requires a decision or action regarding suppression" (Scottish

Government, 2013). We have moved this definition earlier in the manuscript and have elaborated on our discussion of the importance of small vegetation fires and the UK situation in the introduction section to clarify what it is we are interested in and why it is relevant very early on.

Similarly, we have not discriminated on the landcover on which the vegetation fire burned because fires within urban vegetated areas pose a significant threat to people and property. Sadly, the recent fires in California are testament to the importance of this now more than ever. Again, we have placed greater emphasis on why these fires are important in the introduction.

Rather than removing all small fires and specific landcovers, we instead explored these relationships within the manuscript, demonstrating how relationships with PPAs change when using different thresholds of vegetation fire size (using all fires, > 1 ha, > 10 ha, > 50 ha, > 100 ha, > 500 ha) annually and by month. We also look at the number of fires in each of these categories to be very clear about the sample size when considering different thresholds. Section 3.5 (previously section 3.4) also breaks down the relationships according to key landcover types, which excludes fires in urban areas and focuses on heathland/moorland, grassland, broadleaf forest, standing crops, and conifer forest – again providing all details on the number of fires in each category.

Within this review, Reviewer 2 suggested fires smaller than 10 ha, 30 ha or even 50 ha should be excluded, while Reviewer 3 suggested thresholds of 100 ha or 500 ha, which illustrates that applying a size threshold would be an arbitrary decision that there is not a consensus on. We believe that our analysis, which presents the results across key landcovers, months, and for different threshold sizes provides a much more thorough examination of the relationship between PPAs and vegetation fires than narrowing the scope to a single arbitrary size or landcover category. This in turn allows the reader to understand the nuances between fire size and number of fires in the data set when drawing conclusions.

Thank you for entertaining this somewhat lengthy comment, but we hope that this clarifies the decisions made in the manuscript and resolves any concerns the reviewers previously had. We appreciate the chance to improve our manuscript to more clearly articulate these points to the readership of NHESS. We have added a section on UK vegetation fires in the introduction to outline the vegetation fire context, define vegetation fires and introduce the significance of small fires:

"1 Introduction

1.1 Vegetation fire risk in emerging fire prone regions

Vegetation fire risk is increasing in temperate regions like the UK, which have historically experienced few large fires, due to mild, humid climates that mean fuels are generally less flammable (Belcher et al., 2021). However, changes in land management practices combined with a warming climate are increasing the quantity of live biomass available to burn (Glaves et al., 2020). Research in these so-called "emerging fire-prone" regions has been limited because wildfires tend to be dominated by smaller fires, many of which are not detected in satellite records nor included in historical records (Fernandez-Anez et al., 2021). Consequently, decision support systems based on fire weather–fire relationships have primarily been developed in fire-prone regions that have a long history of experiencing large and extreme wildfires (e.g., Canada, Southern Europe, Australia, USA) and then adapted for use in emerging fire-prone regions (e.g., de Jong et al. 2016; Masinda et al. 2022; Steinfeld et al. 2022). Despite and because of these challenges, there is an urgent need to understand and quantify wildfire risk to inform long-term wildfire preparedness in these regions (Pandey et al., 2023).

1.2 Vegetation fires in the UK

Vegetation fires are a semi-natural hazard in the UK as ignitions are almost entirely anthropogenic (Gazzard *et al.* 2016). Human use of fire on the landscape has been a traditional practice for centuries in the UK, particularly as a tool for land management and habitat creation, and fire can bring positive ecological benefits (Belcher et al., 2021); however, there is evidence that the risk of severe vegetation fires is increasing (Arnell et al., 2021; Belcher et al., 2021; Perry et al., 2022). In the UK, Fire and Rescue Services Wildfire Operational Guidance defines a vegetation fire incident as 'any uncontrolled vegetation fire that requires a decision or action regarding suppression' (Scottish Government, 2013). Currently, this does not impose a minimum size threshold on the definition of a vegetation fire, and indeed wildfires do not need to be large to be impactful (Belcher et al., 2021; Kirkland et al., 2023; Stoof et al., 2024). The UK has a high population density of 280 people per square km, compared to traditionally fire prone countries like Australia (3 people per square km), Canada (4 people per square km) and USA (36 people per square km) (World Bank, 2022). This means that natural landscapes are highly fragmented and lack the fuel continuity to generate massive burned areas, and fires tend to be detected quickly. Moreover, a high population density means that a high proportion of fires occur in the interface between people, infrastructure and environment. Vegetation fires in these areas can threaten lives and property, despite their often small size (Graham et al., 2020; John and Rein, 2024; London Fire Brigade, 2023). Critically, vegetation fire preparedness still lags behind other countries and response capabilities can be overwhelmed in extreme conditions (John and Rein, 2024; Climate Change Position Statement, 2025; Pandey et al., 2023).

1.5 Rationale

Fewer studies have explored the large-scale atmospheric drivers of fire weather and vegetation fire occurrence in countries historically not prone to fires, such as the UK. Although fire risk is increasing in these regions, established tools for predicting fire weather and fire occurrence are often lacking. In the UK, extended periods of atmospheric blocking likely play a key role in sufficiently drying out vegetation and elevating fire risk. To address this gap, it is crucial to better understand the role of PPAs in vegetation fires, particularly in temperate regions like the UK, where fires are often smaller than those detectable by satellite but still impactful. A comprehensive fire occurrence database allows us to examine the seasonality and land cover-dependent relationships between PPAs and vegetation fires, extending beyond the larger fires typically observed in summer on specific land covers. We investigate the importance of PPAs for fire weather and vegetation fire occurrence by addressing the following research questions: (1) what is the association between PPAs and surface fire weather across the UK between March–October 2001–2021? Then, using the comprehensive vegetation fire occurrence database for England, (2) What is the association between PPAs and vegetation fire occurrence across key land cover types in England from March–October 2010–2020?"

Major suggestions/comments:

Introduction:

The subsections in the Introduction Section resemble a pile of tiles and make the reading very segmented, descriptive and didactic. I suggest rewriting the introduction in a logical order, avoiding repetitions. A possible solution to avoid a very large section would be the addition of a new section named for instance "Rationale"

*Thank you for this suggestion, we have substantially reworked the introduction on the feedback of all the reviewers and have followed a more logical subheading order and removed any repetition:

"1. Introduction

1.1 Vegetation fire risk in emerging fire prone regions

Vegetation fire risk is increasing in temperate regions like the UK, which have historically experienced few large fires, due to mild, humid climates that mean fuels are generally less flammable (Belcher et al., 2021). However, changes in land management practices combined with a warming climate are increasing the quantity of live biomass available to burn (Glaves et al., 2020). Research in these so-called "emerging fire-prone" regions has been limited because wildfires tend to be dominated by smaller fires, many of which are not detected in satellite records nor included in historical records (Fernandez-Anez et al., 2021). Consequently, decision support systems based on fire weather–fire relationships have primarily been developed in fire-prone regions that have a long history of experiencing large and extreme wildfires (e.g., Canada, Southern Europe, Australia, USA) and then adapted for use in emerging fire-prone regions (e.g., de Jong *et al.* 2016; Masinda *et al.* 2022; Steinfeld *et al.* 2022). Despite and because of these challenges, there is an urgent need to understand and quantify wildfire risk to inform long-term wildfire preparedness in these regions (Pandey et al., 2023).

1.2 Vegetation fires in the UK

Vegetation fires are a semi-natural hazard in the UK as ignitions are almost entirely anthropogenic (Gazzard *et al.* 2016). Human use of fire on the landscape has been a traditional practice for centuries in the UK, particularly as a tool for land management and habitat creation, and fire can bring positive ecological benefits (Belcher et al., 2021); however, there is evidence that the risk of severe vegetation fires is increasing (Arnell et al., 2021; Belcher et al., 2021; Perry et al., 2022). In the UK, Fire and Rescue Services Wildfire Operational Guidance defines a vegetation fire incident as 'any uncontrolled vegetation fire that requires a decision or action regarding suppression' (Scottish Government, 2013). Currently, this does not impose a minimum size threshold on the definition of a vegetation fire, and indeed wildfires do not need to be large to be impactful (Belcher et al., 2021; Kirkland et al., 2023; Stoof et al., 2024). The UK has a high population density of 280 people per square km, compared to traditionally fire prone countries like Australia (3 people per square km), Canada (4 people per square km) and USA (36 people per square km) (World Bank, 2022). This means that natural landscapes are highly fragmented and lack the fuel continuity to generate massive burned areas, and fires tend to be detected quickly. Moreover, a high population density means that a high proportion of fires occur in the interface between people, infrastructure and environment. Vegetation fires in these areas can threaten lives and property, despite their often small size (Graham et al., 2020; John and Rein, 2024; London Fire Brigade, 2023). Critically, vegetation fire preparedness still lags behind other countries and response capabilities can be overwhelmed in extreme conditions (John and Rein, 2024; Climate Change Position Statement, 2025; Pandey et al., 2023).

1.3 Synoptic controls on surface fire weather and vegetation fire

In the midlatitudes, surface weather is driven by synoptic-scale weather patterns (i.e., large scale upper-air atmospheric circulation patterns (Franzke et al., 2020)). While surface weather is highly spatiotemporally variable and difficult to forecast beyond the short-term, synoptic-scale upper-air (500 hPa) atmospheric patterns can be more reliably predicted in the medium range (+10 days) (Hohenegger and Schär, 2007). As such, considering synoptic-scale indicators of vegetation fire occurrence in addition to surface fire weather may provide additional insights for improving near-to-medium range forecasting of fire danger to aid fire preparedness and management decision-making (Humphrey et al., 2024; Jain et al., 2024; Papavasileiou and Giannaros, 2023).

Within Europe, previous research examining large-scale weather patterns associated with vegetation fire occurrence have tended to focus on countries in the Mediterranean (e.g., Duane and Brotons 2018; Pineda *et al.* 2022; Rodrigues *et al.* 2022), likely due to the history of significant vegetation fires and comprehensive fire occurrence databases; though there are some exceptions that examine Europe-wide (Giannaros and Papavasileiou, 2023; Little et al., 2024) and Northern European (Drobyshev et al., 2021; Wastl et al., 2013) relationships. While atmospheric blocking has been associated with vegetation fire

occurrence across Southern Europe, other studies have also highlighted the importance of strong wind events and atmospheric instability as drivers of extreme vegetation fire activity (Artés et al., 2022; Resco de Dios et al., 2022; Ruffault et al., 2017).

1.4 Persistent positive anomalies in geopotential heights (PPAs)

Atmospheric blocking occurs when a high pressure system remains nearly stationary such that it effectively "blocks" the usual mid-latitude zonal airflow, leading to dry, clear-sky conditions and high surface temperatures that may be amplified by land–atmosphere feedbacks (Rex, 1950). Such conditions promote fuel aridity and consequently vegetation fire occurrence (Sharma et al., 2022). Moreover, persistent atmospheric blocking events can lead to synchronous elevated fire danger across large areas, which can overwhelm fire response capabilities (Abatzoglou et al., 2021; Jain et al., 2024).

Positive geopotential height anomalies at the 500 hPa level are widely used to identify high-pressure blocking events (Tibaldi and Molteni, 1990). One such method, Persistent Positive Anomalies in 500 hPa geopotential heights (PPAs), are an event-based paradigm for tracking extremes in high pressure blocking patterns (that exceed a threshold amplitude, size, and duration) through space and time (Dole and Gordon 1983). Compared with other methods of identifying atmospheric blocking patterns, such as measuring the reversal of meridional flow (Pinheiro et al., 2019; Tibaldi and Molteni, 1990) and dynamic potential vorticity (Pelly and Hoskins, 2003; Small et al., 2014), PPAs are less constrained by the specific blocking mechanism, which can be important for capturing events during the main fire season when strong polar dynamics are not common (Dole and Gordon, 1983; Miller et al., 2020). PPAs can be used to capture the potential persistent, weaker pressure gradient events that characterise hot, dry surface conditions for fires in summer (Sousa et al., 2018; Woollings et al., 2018).

PPAs have recently been associated with extreme fire weather (hot, dry weather as defined by the Canadian Fire Weather Index) and vegetation fires in the Northern Hemisphere mid-latitudes for both Western North America (Jain et al., 2024; Sharma et al., 2022) and Europe (Little et al., 2024). We recently established the importance of PPAs for vegetation fires at a pan-European scale finding that fires were more than twice as likely to occur during PPA events across Europe and were associated with 53% of burned area for Western Europe (Little et al., 2024). However, the EFFIS burned area product used in that study only includes vegetation fires of around 30 ha and greater detected by satellite imagery, which resulted in very few records in regions with predominantly small vegetation fires. Notably, for the period March–October 2010–2020, EFFIS reported 348 vegetation fires for the UK (San-Miguel-Ayanz et al., 2012). In comparison, for the same period, the Fire and Rescue Service incident database reported 291,963 vegetation fires occurring in England alone (Forestry Commission, 2023).

1.5 Rationale

Fewer studies have explored the large-scale atmospheric drivers of fire weather and vegetation fire occurrence in countries historically not prone to fires, such as the UK. Although fire risk is increasing in these regions, established tools for predicting fire weather and fire occurrence are often lacking. In the UK, extended periods of atmospheric blocking likely play a key role in sufficiently drying out vegetation and elevating fire risk. To address this gap, it is crucial to better understand the role of PPAs in vegetation fires, particularly in temperate regions like the UK, where fires are often smaller than those detectable by satellite but still impactful. A comprehensive fire occurrence database allows us to examine the seasonality and land cover-dependent relationships between PPAs and vegetation fires, extending beyond the larger fires typically observed in summer on specific land covers.   We investigate the importance of PPAs for fire weather and vegetation fire occurrence by addressing the following research questions: (1) what is the association between PPAs and surface fire weather across the UK between March–October 2001–2021? Then, using the comprehensive vegetation fire occurrence database for England, (2) What is the association between PPAs and vegetation fire occurrence across key land cover types in England from March–October 2010–2020?"

Another important topic related to the scope of this paper is the definition of wildfire and the use of this term in case of small fires not detectable by satellite. Rewrite the manuscript having this in mind. (please see the comments below).

*We trust this question has been thoroughly resolved in the overall comment to Reviewer 2. To improve the clarity and focus of the manuscript and avoid entering into debate around terminology, we have opted to remove the term 'wildfire' from the manuscript and instead refer to 'vegetation fire occurrence'. This highlights both that we refer to outdoor fires where vegetation is burning and that it is the occurrence of a fire (rather than the subsequent behaviour of the fire) that we are interested in. We have added a section to the introduction to explain the importance of small vegetation fires and to define vegetation fires in the UK context.

Still important is the confusion between vegetation cover and fuel load and moisture, as well as the concept and quantification of ignitions.

*We are not sure what part of the text this comment refers to, but we have taken care to ensure all terminology is correct and clearly explains the linkages between concepts. For example, in lines 38–39 and line 392 we have now specified that we are referring to live fuels (rather than just fuels).

Line 37- The closed link between heat and dry extremes and low fuel content is known. Please clarify how heatwaves and droughts are causing increased fuel availability

*We were referring to how existing fuel that previously has been too wet to burn (see previous sentence) is drying out and becoming available fuel for vegetation fires (rather than an increase in net biomass over the long term). We can see how the phrase fuel availability may be misleading so have changed the wording to be less ambiguous:

"Vegetation fire risk is increasing in temperate regions like the UK, which have historically experienced few large fires, due to mild, humid climates that mean fuels are generally less flammable (Belcher et al., 2021). However, changes in land management practices combined with a warming climate are increasing the quantity of live biomass available to burn (Glaves et al., 2020)."

Lines 100-101 – Accordingly, with the fire triangle on Moritz et al., (2005) the wildfires are driven by weather, fuel and topography and this rationale is based on the temporal and spatial extension of fires and weather or synoptic patterns must be linked to a fire with temporal and spatial extension that surpass a single ignition (e.g., domestic or garden fires).

*Apologies, we do not understand how this comment relates to this section of the text, are these the correct line numbers? We do not look to only consider single ignitions, rather the cumulative burned area and elevated number of ignitions under PPA conditions. A large number of small ignitions would meet this extension. The purpose of this section of the text is to introduce the novelty of this comprehensive dataset and the significance of small fires in this region, which is a core reason for conducting this research. We do wonder if maybe the reviewer thought we were looking at the role of PPAs through the life time of a fire, i.e., the impact on fire behaviour? If this is the case, we have clarified this in the text by replacing the term 'wildfire' with 'vegetation fire occurrence'.

Such a large number of fires could be classified as wildfires? How many burned area are associated with this large number of fires? What were the causes of such fires?

*As discussed above, changing the definition of a wildfire used in the UK or trying to form an international consensus on the definition of a wildfire is well outside the aim and scope of this paper and would be better suited to an opinion piece style publication (which would be interesting!). Instead, we have opted to remove the term 'wildfire' from this manuscript to avoid entering into this discussion and replaced it with 'vegetation fire occurrences', which is more specific and less associated with perceived definitions the reader may have (e.g., Tedim and Leone, 2020). We have also added text to the introduction to explain early on the definition of a vegetation fire and the context around small fires in the UK, backed by other references. The causes of vegetation fires in the UK are not well known and there are many gaps and errors in the cause attributed to incidents in this dataset so it would not be suitable to analyse cause here. Nearly all ignitions in the UK are anthropogenic in origin in one form or another, which we have also elaborated on in the introduction. Please see the overall comment to Reviewer 2 for changes we have made to the revised manuscript to address your concerns.

Lines 102-103 – The fires not detectable by satellite data are forest, rural or crop fires? Are they classified as wildfires? Which kind of impacts?

*We trust this comment has been well addressed elsewhere in the response and through the added paragraph on small fires in the introduction. Please see the overall comment to reviewer 2 for changes we have made to the revised manuscript to address your concerns.

Lines 119-123 – Move to the introduction section

*We have moved this to the introduction section as suggested.

Figure 1 - Present in the figure the labels (a), (b), (c), and (d).

*Done.

Figure 1 shows an enormous number of fires burned less than 1 ha (almost 3000 in Spring and Summer), which does not have representation in terms of burned area (figures on left panel). Consider replacing the figure with a new figure using the threshold of 50ha or 100ha.

*We have updated this figure following other reviewer suggestions and have log transformed the y axes for readability rather than applying a threshold. This has now become Figure 4 in the results section. We have also added Figure S5 to the Supplementary Material, which shows the initial figure without log transformation as well as using the threshold of 50 ha in a 4 panel plot.

[Figure]

"Figure 4: Total log transformed daily (a) burned area and (b) number of vegetation fires recorded in England between 2010–2020. Day of year on the x-axis is partitioned to show the calendar months (dashed lines) winter (DJF), spring (MAM), summer (JJA) and autumn (SON)."

[Figure]

"Figure S5: Total daily (a) burned area, (b) number of wildfires, (c) burned area from vegetation fires at least 50 ha in size and (d) number of fires at least 50 ha in size recorded in England on each day of the year between 2010–2020."

Data:

Lines 164-174 – Move to introduction section.

*We agree with the reviewer and have moved this to the introduction to elaborate on PPAs.

Lines 204-210: How were the individual fire events aggregated to each 1 x1 grid cell?

*We apologise; this was an analysis that was not included in the final manuscript so this text should have been removed. The percentage of burned area and fires associated with PPAs was analysed by fire record rather than by grid cell (the opposite to how it was communicated originally). The text now reads:

"We labelled each incident as a PPA–fire event if a PPA was present in the grid cell the incident occurred in either on the same day or up to five days preceding the incident, otherwise it was labelled a noPPA–fire event. This five day lag is to account for the role of PPA conditions in pre-drying fuels that subsequently ignite (supported by Fig. S1 and previous research (Sharma et al., 2022)). (Little et al., 2024) assessed the sensitivity of PPA–fire associations across Europe to a range of different time lags, finding no major differences in the results. It should be noted that incident burned area is assigned to the incident start date as there is no daily breakdown of burned area within individual events. We acknowledge this limitation; however, as 99.9% of all vegetation fires and 72.7% of burned area are from incidents that occur within a single day or up to five days length, we believe this metric still largely captures whether incidents are associated with PPA events (particularly as PPA–fire events are defined by a five-day lag period). We also repeated the above steps but first applying a minimum fire size threshold of greater than 1 ha; 10 ha; 50 ha; 100 ha; and 500 ha and filtering by fires on specific key UK landcovers (heathland/moorland; grassland (grassland, pasture, grazing); conifer forest; broadleaf forest; and standing crops) to examine PPA–fire relationships across different vegetation fire sizes and landcover types."

The small fires may have a panoply of drivers. An ignition may have different causes, but fire propagation is related to fuel, weather and topography. To evaluate the role of weather we should have a fire higher than a single ignition and therefore consider only the fires higher than 50ha or 100ha.

*We agree that there are many drivers of small fires, which is why we look at the total amount of burned area and number of fire occurrences associated with PPA events. The records refer to the individual incident, so do not specify whether the incident was the result of a single ignition or multiple ignitions that joined up. By looking at cumulative burned area and number of fire occurrences, we give the capacity to identify days where there were an elevated number of vegetation fires occurring, even if they only burned a small area (which may be related to many other factors such as fuel continuity, suppression actions amongst other reasons). The 50 ha / 100 ha threshold is arbitrary and not appropriate for the study location as discussed above, but we have presented the results using different size thresholds (for both burned area and number of fires) in the main text for the interested reader. We wonder if the reviewer is thinking we are looking at PPAs across the lifetime of wildfires and how they impact wildfire behaviour instead of fire occurrences? If so, we apologise for the lack of clarity here. By replacing the term 'wildfire' with 'vegetation fire occurrence' and being more specific when talking about occurrences we hope we have addressed this issue.

In the case of using statistical analysis be careful with the sample size and significance level.

*We are sorry, but we are not sure what aspect of the text you are referring to. Our assumption is Figure 4, with the linear regression analysis? We can confirm that the sample sizes are all sufficient to conduct this analysis (sufficient PPA versus non PPA days by grid cell and month across the 2001-2021 period) and we report the statistical significance ($p < 0.05$) by grid cell.

Figure 3 – For the sake of clarity, present in figures the name of the variables

*The names of the variables are on the right hand axes of the figures, but we have also labelled them a-i with the names in the caption also for clarity. This is now Figure 2 in the revised version.

Results

Consider showing how the predictable skill of PPA for fire occurrence is better than climatology or persistence.

*We are sorry, but we are not 100% sure we understand what the reviewer is suggesting here. Are you referring to climatology / persistence in terms of fire occurrence (i.e., can PPAs predict fire occurrence beyond seasonal fire activity) or the climatology/persistence of PPAs / geopotential height anomalies?

Discussion:

Lines 382-392: the role played by PPA on fuel and ignitions is quite qualitative. The authors present in the manuscript an analysis using land cover types. Fuel conditions, such as loads and moisture could not be extrapolated from land cover types.

*Yes, the discussion of the different landcover types burning is qualitative for the reasons you say, that it is not possible to extrapolate fuel loads and moisture in a quantitative manner. We use current understanding of fires in these different landcovers (i.e., how heathland/moorland fires behave and compare to grassland and crop fires) based on published literature and the database that shows when these fires occur to inform our discussion of the role of PPAs on fuel and ignitions. We have been careful to word these as our 'hypotheses' of the relationships, rather than stating these as our results.

Clarify the threshold of fuel moisture to allow ignitions. How is ignition frequency estimated? I suggest including a quantitative analysis of fuel load and moisture analysis or removing such sentences.

*As this is the discussion section, we are simply putting our results in context with previous literature here, but we have taken care in the revised manuscript to be clear about what are our findings and what are hypotheses and we have also been clear with the terminology used.

Lines 401-407: If the authors do not consider FWI adequate for spring fire weather danger, why FWI is used in this analysis? Which kind of tools should be used instead? Please consider clarifying the ideas in the paragraph.

*We include the FWI analysis because it is so widely used and in lieu of a replacement that is so well established, like has been used in many other studies in regions outside of Canada (e.g., Fogarty et al., 1998; Dimitrakopoulos et al., 2011; Wastl et al., 2013, including UK-based studies (e.g., Arnell et al., 2021; de Jong et al., 2016; Perry et al., 2022) and global studies (e.g., Bedia et al., 2015; Abatzoglou et al., 2019)). Furthermore, other subcomponents of the CFWIS, e.g., the FFMC, perform better in spring than the FWI subcomponent, which seems to perform better in

summer. This finding is supported by other research in the UK (Davies and Legg, 2016; Nikonovas et al., 2024) and other temperate regions (Lambrechts et al., 2024), and we have added a section 3.4 in the results that shows the distribution of CFWIS indices during vegetation fires in England between spring and summer (below).

We also realised the wording of the paragraph in lines 401-407 is quite strong but ambiguous, e.g., it may not have been clear that we were referring to the FWI subcomponent rather than the entire CFWIS of the same name. We also state that phenological indicators offer an improvement in spring and point the reader to references that support this. We have clarified these aspects in the text:

"To date, the dual spring and summer fire seasons have been a major challenge for predicting fire weather in the UK as the fuels burning and their drivers differ seasonally (Belcher et al., 2021). Surface fire weather indices like the FWI subcomponent of the CFWIS perform reasonably well in summer but fail to capture the spring fire season (Davies and Legg, 2016; de Jong et al., 2016; Nikonovas et al., 2024), while phenological indicators (for example, vegetation greenness indices like the Enhanced Vegetation Index 2 (EVI2) (Nikonovas et al., 2024)) improve spring fire season predictions but do not capture the heatwave events during summer that can lead to extreme drying of live fuels (Ivison et al., 2024; Nikonovas et al., 2024). It is important to consider all available tools to accurately predict different periods of fire danger, including surface variables like weather, landcover, and phenology, but also synoptic indicators."

"3.4 Surface fire weather and vegetation fire occurrence across England

Surface fire weather, as indicated by the CFWIS indices, tends to be anomalously high when vegetation fires occur in England (Fig. 5). This is true for both spring and summer fires less than and greater than 50 ha, though fire weather anomalies are highest for larger fires in summer. In spring, the FFMC is the most elevated when vegetation fires occur, while the other indices are distributed around the average or slightly above average in the case of the FWI. In summer, the DC and BUI are anomalously high when vegetation fires occur and the FWI is positively skewed for large fires but not fires < 50 ha.

[Figure]

Figure 5: Density plots showing the distribution of the Canadian Fire Weather Index System indices anomalies: (a) FWI, (b) FFMC, (c) DMC, (d) DC, (e) ISI and (f) BUI for spring (left subpanel) and summer (right subpanel). Distributions are presented for wildfires less than or equal to 50 ha (blue fill shows the distribution, dashed blue vertical line shows the mean value) and wildfires greater than 50 ha (green fill shows the distribution, dashed green vertical line shows the mean value). The black vertical line at zero separates positive anomalies (indices are higher than average during wildfires) from negative anomalies (indices are below average during wildfires). Note the independent axes labels for readability."

Lines 428-438: Not clear how the PPA methodology could improve the fire rating in England or how this information will be integrated into it.

*We do not suggest that the PPA methodology be integrated into MOFSI, as there are many other issues with the MOFSI tool, which is fundamentally used for a different purpose. Rather, we believe that synoptic indicators, like PPAs, could be used as an additional tool to provide a longer-range forecast of elevated wildfire danger periods, something that is not currently available in the UK. Once we have a better understanding of fuel moisture thresholds for vegetation fires in the UK across key landcovers, PPA forecasts could be combined with (for example) observations of soil moisture as antecedent soil moisture conditions contribute to heatwaves (Horowitz et al., 2022) or combined with other circulation patterns that PPAs are known to have nonlinear interactions with (Bartusek et al., 2022) to provide a better

understanding of elevated fire weather periods driven by multiple controls. We have added some text to improve the clarity of this section:

"The UK does not have a fire danger rating system although the Met Office Fire Severity Index (MOFSI), an adaptation of the Canadian FWI, is used to trigger land closures in England and Wales during the most extreme conditions (England and Wales Fire Severity Index, 2023). The evolution of PPA events through 500 hPa geopotential height forecasts could be tracked to identify periods of sustained elevated fire weather that may challenge or overwhelm available resources for fire management (Jain et al., 2024). Synoptic indicators of elevated wildfire danger periods would therefore provide significant benefits for near-to-medium range wildfire preparedness and awareness in emerging fire-prone regions like the UK where tailored assessments of fire danger do not yet exist."

**References**

Abatzoglou, J. T., Williams, A. P., and Barbero, R.: Global Emergence of Anthropogenic Climate Change in Fire Weather Indices, Geophysical Research Letters, 46, 326–336, https://doi.org/10.1029/2018GL080959, 2019.

Abatzoglou, J. T., Juang, C. S., Williams, A. P., Kolden, C. A., and Westerling, A. L.: Increasing Synchronous Fire Danger in Forests of the Western United States, Geophysical Research Letters, 48, e2020GL091377, https://doi.org/10.1029/2020GL091377, 2021.

Arnell, N. W., Freeman, A., and Gazzard, R.: The effect of climate change on indicators of fire danger in the UK, Environ. Res. Lett., 16, 044027, https://doi.org/10.1088/1748-9326/abd9f2, 2021.

Artés, T., Castellnou, M., Houston Durrant, T., and San-Miguel, J.: Wildfire–atmosphere interaction index for extreme-fire behaviour, Natural Hazards and Earth System Sciences, 22, 509–522, https://doi.org/10.5194/nhess-22-509-2022, 2022.

Bartusek, S., Kornhuber, K., and Ting, M.: 2021 North American heatwave amplified by climate change-driven nonlinear interactions, Nat. Clim. Chang., 12, 1143–1150, https://doi.org/10.1038/s41558-022-01520-4, 2022.

Bedia, J., Herrera, S., Gutiérrez, J. M., Benali, A., Brands, S., Mota, B., and Moreno, J. M.: Global patterns in the sensitivity of burned area to fire-weather: Implications for climate change, Agricultural and Forest Meteorology, 214–215, 369–379, https://doi.org/10.1016/j.agrformet.2015.09.002, 2015.

Belcher, C. M., Brown, I., Clay, G. D., Doerr, S. H., Elliott, A., Gazzard, R., Kettridge, N., Morison, J., Perry, M., and Smith, T. E. L.: UK Wildfires and their Climate Challenges, 2021.

Davies, G. M. and Legg, C. J.: Regional variation in fire weather controls the reported occurrence of Scottish wildfires, PeerJ, 4, e2649, https://doi.org/10.7717/peerj.2649, 2016.

Dimitrakopoulos, A. P., Bemmerzouk, A. M., and Mitsopoulos, I. D.: Evaluation of the Canadian fire weather index system in an eastern Mediterranean environment, Meteorological Applications, 18, 83–93, https://doi.org/10.1002/met.214, 2011.

Dole, R. M. and Gordon, N. D.: Persistent Anomalies of the Extratropical Northern Hemisphere Wintertime Circulation: Geographical Distribution and Regional Persistence Characteristics,

Monthly Weather Review, 111, 1567–1586, https://doi.org/10.1175/1520-0493(1983)111<1567:PAOTEN>2.0.CO;2, 1983.

Drobyshev, I., Ryzhkova, N., Eden, J., Kitenberga, M., Pinto, G., Lindberg, H., Krikken, F., Yermokhin, M., Bergeron, Y., and Kryshen, A.: Trends and patterns in annually burned forest areas and fire weather across the European boreal zone in the 20th and early 21st centuries, Agricultural and Forest Meteorology, 306, 108467, https://doi.org/10.1016/j.agrformet.2021.108467, 2021.

Duane, A. and Brotons, L.: Synoptic weather conditions and changing fire regimes in a Mediterranean environment, Agricultural and Forest Meteorology, 253–254, 190–202, https://doi.org/10.1016/j.agrformet.2018.02.014, 2018.

Fernandez-Anez, N., Krasovskiy, A., Müller, M., Vacik, H., Baetens, J., Hukić, E., Kapovic Solomun, M., Atanassova, I., Glushkova, M., Bogunović, I., Fajković, H., Djuma, H., Boustras, G., Adámek, M., Devetter, M., Hrabalikova, M., Huska, D., Martínez Barroso, P., Vaverková, M. D., Zumr, D., Jõgiste, K., Metslaid, M., Koster, K., Köster, E., Pumpanen, J., Ribeiro-Kumara, C., Di Prima, S., Pastor, A., Rumpel, C., Seeger, M., Daliakopoulos, I., Daskalakou, E., Koutroulis, A., Papadopoulou, M. P., Stampoulidis, K., Xanthopoulos, G., Aszalós, R., Balázs, D., Kertész, M., Valkó, O., Finger, D. C., Thorsteinsson, T., Till, J., Bajocco, S., Gelsomino, A., Amodio, A. M., Novara, A., Salvati, L., Telesca, L., Ursino, N., Jansons, A., Kitenberga, M., Stivrins, N., Brazaitis, G., Marozas, V., Cojocaru, O., Gumeniuc, I., Sfecla, V., Imeson, A., Veraverbeke, S., Mikalsen, R. F., Koda, E., Osinski, P., Castro, A. C. M., Nunes, J. P., Oom, D., Vieira, D., Rusu, T., Bojović, S., Djordjevic, D., Popovic, Z., Protic, M., Sakan, S., Glasa, J., Kacikova, D., Lichner, L., Majlingova, A., Vido, J., Ferk, M., Tičar, J., Zorn, M., Zupanc, V., Hinojosa, M. B., Knicker, H., Lucas-Borja, M. E., Pausas, J., Prat-Guitart, N., Ubeda, X., Vilar, L., Destouni, G., Ghajarnia, N., Kalantari, Z., Seifollahi-Aghmiuni, S., Dindaroglu, T., Yakupoglu, T., Smith, T., Doerr, S., and Cerda, A.: Current Wildland Fire Patterns and Challenges in Europe: A Synthesis of National Perspectives, Air, Soil and Water Research, 14, 11786221211028185, https://doi.org/10.1177/11786221211028185, 2021.

Flannigan, M. D., Wotton, B. M., Marshall, G. A., de Groot, W. J., Johnston, J., Jurko, N., and Cantin, A. S.: Fuel moisture sensitivity to temperature and precipitation: climate change implications, Climatic Change, 134, 59–71, https://doi.org/10.1007/s10584-015-1521-0, 2016.

Fogarty, L. G., Pearce, G., Catchpole, W. R., and Alexander, M. E.: Adoption vs. adaptation: lessons from applying the Canadian forest fire danger rating system in New Zealand, Proceedings, 3rd International Conference on Forest Fire Research and 14th Fire and Forest Meteorology Conference, Luso, Coimbra, Portugal, 1011–1028, 1998.

Forestry Commission: Wildfire statistics for England: Report to 2020-21, Forestry Commission England, Bristol, 2023.

Franzke, C. L. E., Barbosa, S., Blender, R., Fredriksen, H.-B., Laepple, T., Lambert, F., Nilsen, T., Rypdal, K., Rypdal, M., Scotto, Manuel G., Vannitsem, S., Watkins, N. W., Yang, L., and Yuan, N.: The Structure of Climate Variability Across Scales, Reviews of Geophysics, 58, e2019RG000657, https://doi.org/10.1029/2019RG000657, 2020.

Gazzard, R., McMorrow, J., and Aylen, J.: Wildfire policy and management in England: an evolving response from Fire and Rescue Services, forestry and cross-sector groups, Philos Trans R Soc Lond B Biol Sci, 371, 20150341, https://doi.org/10.1098/rstb.2015.0341, 2016.

Giannaros, T. M. and Papavasileiou, G.: Changes in European fire weather extremes and related atmospheric drivers, Agricultural and Forest Meteorology, 342, 109749, https://doi.org/10.1016/j.agrformet.2023.109749, 2023.

Glaves, D. J., Crowle, A. J., Bruemmer, C., and Lenaghan, S. A.: The causes and prevention of wildfire on heathlands and peatlands in England (NEER014), Peterborough, 2020.

Graham, A. M., Pope, R. J., Pringle, K. P., Arnold, S., Chipperfield, M. P., Conibear, L. A., Butt, E. W., Kiely, L., Knote, C., and McQuaid, J. B.: Impact on air quality and health due to the Saddleworth Moor fire in northern England, Environ. Res. Lett., 15, 074018, https://doi.org/10.1088/1748-9326/ab8496, 2020.

Hohenegger, C. and Schär, C.: Atmospheric Predictability at Synoptic Versus Cloud-Resolving Scales, Bulletin of the American Meteorological Society, 88, 1783–1793, 2007.

Horowitz, R. L., McKinnon, K. A., and Simpson, I. R.: Circulation and Soil Moisture Contributions to Heatwaves in the United States, https://doi.org/10.1175/JCLI-D-21-0156.1, 2022.

Humphrey, R., Saltenberger, J., Abatzoglou, J. T., and Cullen, A.: Near-term fire weather forecasting in the Pacific Northwest using 500-hPa map types, Int. J. Wildland Fire, 33, https://doi.org/10.1071/WF23117, 2024.

Ivison, K., Little, K., Orpin, A., Lewis, C. H. M., Dyer, N., Keyzor, L., Everett, L., Stoll, E., Andersen, R., Graham, L. J., and Kettridge, N.: A national-scale sampled temperate fuel moisture database, Sci Data, 11, 973, https://doi.org/10.1038/s41597-024-03832-w, 2024.

Jain, P., Sharma, A. R., Acuna, D. C., Abatzoglou, J. T., and Flannigan, M.: Record-breaking fire weather in North America in 2021 was initiated by the Pacific northwest heat dome, Commun Earth Environ, 5, 1–10, https://doi.org/10.1038/s43247-024-01346-2, 2024.

John, J. and Rein, G.: Heatwaves and firewaves: the drivers of urban wildfires in London in the summer of 2022, https://doi.org/10.21203/rs.3.rs-4774726/v1, 23 July 2024.

de Jong, M. C., Wooster, M. J., Kitchen, K., Manley, C., Gazzard, R., and McCall, F. F.: Calibration and evaluation of the Canadian Forest Fire Weather Index (FWI) System for improved wildland fire danger rating in the United Kingdom, Natural Hazards and Earth System Sciences, 16, 1217–1237, https://doi.org/10.5194/nhess-16-1217-2016, 2016.

Kirkland, M., Atkinson, P. W., Pearce-Higgins, J. W., de Jong, M. C., Dowling, T. P. F., Grummo, D., Critchley, M., and Ashton-Butt, A.: Landscape fires disproportionally affect high conservation value temperate peatlands, meadows, and deciduous forests, but only under low moisture conditions, Science of The Total Environment, 884, 163849, https://doi.org/10.1016/j.scitotenv.2023.163849, 2023.

Lambrechts, H. A., Stoof, C. R., del Pozo, M., Ludwig, F., and Paparrizos, S.: The role of weather and climate information services to support in wildfire management in Northwestern Europe, Climate Risk Management, 46, 100672, https://doi.org/10.1016/j.crm.2024.100672, 2024.

Little, K., Castellanos-Acuna, D., Jain, P., Graham, L. J., Kettridge, N., and Flannigan, M.: Persistent positive anomalies in geopotential heights drive enhanced wildfire activity across Europe, Philosophical Transactions of the Royal Society B: Biological Sciences, https://doi.org/10.1098/rstb.2023.0455, 2024.

Liu, P., Zhu, Y., Zhang, Q., Gottschalck, J., Zhang, M., Melhauser, C., Li, W., Guan, H., Zhou, X., Hou, D., Peña, M., Wu, G., Liu, Y., Zhou, L., He, B., Hu, W., and Sukhdeo, R.: Climatology of tracked persistent maxima of 500-hPa geopotential height, Clim Dyn, 51, 701–717, https://doi.org/10.1007/s00382-017-3950-0, 2018.

London Fire Brigade: Major Incident Review Extreme Weather Period 2022, 2023.

Masinda, M. M., Li, F., Qi, L., Sun, L., and Hu, T.: Forest fire risk estimation in a typical temperate forest in Northeastern China using the Canadian forest fire weather index: case study in autumn 2019 and 2020, Nat Hazards, 111, 1085–1101, https://doi.org/10.1007/s11069-021-05054-4, 2022.

McElhinny, M., Beckers, J. F., Hanes, C., Flannigan, M., and Jain, P.: A high-resolution reanalysis of global fire weather from 1979 to 2018 - Overwintering the Drought Code, Earth System Science Data, 12, 1823–1833, https://doi.org/10.5194/essd-12-1823-2020, 2020.

England and Wales Fire Severity Index: https://www.metoffice.gov.uk/public/weather/fire-severity-index, last access: 25 April 2023.

Miller, R. L., Lackmann, G. M., and Robinson, W. A.: A New Variable-Threshold Persistent Anomaly Index: Northern Hemisphere Anomalies in the ERA-Interim Reanalysis, Monthly Weather Review, 148, 43–62, https://doi.org/10.1175/MWR-D-19-0144.1, 2020.

Climate Change Position Statement: https://nfcc.org.uk/our-services/position-statements/climate-change-position-statement/, last access: 10 February 2025.

Nikonovas, T., Santín, C., Belcher, C. M., Clay, G. D., Kettridge, N., Smith, T. E. L., and Doerr, S. H.: Vegetation phenology as a key driver for fire occurrence in the UK and comparable humid temperate regions, Int. J. Wildland Fire, 33, https://doi.org/10.1071/WF23205, 2024.

Pandey, P., Huidobro, G., Lopes, L. F., Ganteaume, A., Ascoli, D., Colaco, C., Xanthopoulos, G., Giannaropoulos, T., Gazzard, R., Boustras, G., Steelman, T., Charlton, V., Ferguson, E., Kirschner, J., Little, K., Stoof, C., Nikolakis, W., Fernández-Blanco, C. R., Ribotta, C., Lambrechts, H., Fernandez, M., and Dossi, S.: A global outlook on increasing wildfire risk: current policy situation and future pathways, Trees, Forests and People, 100431, https://doi.org/10.1016/j.tfp.2023.100431, 2023.

Papavasileiou, G. and Giannaros, T. M.: The Predictability of the Synoptic-Scale Fire Weather Conditions during the 2018 Mati Wildfire, Environmental Sciences Proceedings, 26, 164, https://doi.org/10.3390/environsciproc2023026164, 2023.

Pelly, J. L. and Hoskins, B. J.: A New Perspective on Blocking, 2003.

Perry, M. C., Vanvyve, E., Betts, R. A., and Palin, E. J.: Past and future trends in fire weather for the UK, Natural Hazards and Earth System Sciences, 22, 559–575, https://doi.org/10.5194/nhess-22-559-2022, 2022.

Pineda, N., Peña, J. C., Soler, X., Aran, M., and Pérez-Zanón, N.: Synoptic weather patterns conducive to lightning-ignited wildfires in Catalonia, in: Advances in Science and Research, 21st EMS Annual Meeting - virtual: European Conference for Applied Meteorology and Climatology 2021 -, 39–49, https://doi.org/10.5194/asr-19-39-2022, 2022.

Pinheiro, M. C., Ullrich, P. A., and Grotjahn, R.: Atmospheric blocking and intercomparison of objective detection methods: flow field characteristics, Clim Dyn, 53, 4189–4216, https://doi.org/10.1007/s00382-019-04782-5, 2019.

Resco de Dios, V., Cunill Camprubí, À., Pérez-Zanón, N., Peña, J. C., Martínez del Castillo, E., Rodrigues, M., Yao, Y., Yebra, M., Vega-García, C., and Boer, M. M.: Convergence in critical fuel moisture and fire weather thresholds associated with fire activity in the pyroregions of Mediterranean Europe, Science of The Total Environment, 806, 151462, https://doi.org/10.1016/j.scitotenv.2021.151462, 2022.

Rex, D. F.: Blocking Action in the Middle Troposphere and its Effect upon Regional Climate, Tellus, 2, 196–211, https://doi.org/10.1111/j.2153-3490.1950.tb00331.x, 1950.

Rodrigues, M., Camprubí, À. C., Balaguer-Romano, R., Ruffault, J., Fernandes, P. M., and Dios, V. R. de: Drivers and implications of the extreme 2022 wildfire season in Southwest Europe, https://doi.org/10.1101/2022.09.29.510113, 30 September 2022.

Ruffault, J., Moron, V., Trigo, R. M., and Curt, T.: Daily synoptic conditions associated with large fire occurrence in Mediterranean France: evidence for a wind-driven fire regime, International Journal of Climatology, 37, 524–533, https://doi.org/10.1002/joc.4680, 2017.

San-Miguel-Ayanz, J., Schulte, E., Schmuck, G., Camia, A., Strobl, P., Liberta, G., Giovando, C., Boca, R., Sedano, F., Kempeneers, P., McInerney, D., Withmore, C., de Oliveira, S. S., Rodrigues, M., Durrant, T., Corti, P., Oehler, F., Vilar, L., and Amatulli, G.: Comprehensive Monitoring of Wildfires in Europe: The European Forest Fire Information System (EFFIS), in: Approaches to Managing Disaster - Assessing Hazards, Emergencies and Disaster Impacts, edited by: Tiefenbacher, J., InTech, https://doi.org/10.5772/28441, 2012.

Scottish Government: Fire and Rescue Service Wildfire Operational Guidance, Scottish Government, Edinburgh, 2013.

Sharma, A. R., Jain, P., Abatzoglou, J. T., and Flannigan, M.: Persistent Positive Anomalies in Geopotential Heights Promote Wildfires in Western North America, Journal of Climate, 35, 2867–2884, https://doi.org/10.1175/JCLI-D-21-0926.1, 2022.

Small, D., Atallah, E., and Gyakum, J. R.: An Objectively Determined Blocking Index and its Northern Hemisphere Climatology, Journal of Climate, 27, 2948–2970, 2014.

Sousa, P. M., Trigo, R. M., Barriopedro, D., Soares, P. M. M., and Santos, J. A.: European temperature responses to blocking and ridge regional patterns, Clim Dyn, 50, 457–477, https://doi.org/10.1007/s00382-017-3620-2, 2018.

Steinfeld, D., Peter, A., Martius, O., and Brönnimann, S.: Assessing the performance of various fire weather indices for wildfire occurrence in Northern Switzerland, EGUsphere, 1–23, https://doi.org/10.5194/egusphere-2022-92, 2022.

Stoof, C. R., Kok, E., Cardil Forradellas, A., and van Marle, M. J. E.: In temperate Europe, fire is already here: The case of The Netherlands, Ambio, https://doi.org/10.1007/s13280-023-01960-y, 2024.

Tedim, F. and Leone, V.: The Dilemma of Wildfire Definition: What It Reveals and What It Implies, Front. For. Glob. Change, 3, https://doi.org/10.3389/ffgc.2020.553116, 2020.

Tibaldi, S. and Molteni, F.: On the operational predictability of blocking, Tellus A: Dynamic Meteorology and Oceanography, 42, 343–365, https://doi.org/10.3402/tellusa.v42i3.11882, 1990.

Van Wagner, C. E.: Development and structure of the Canadian forest fire weather index system, 37 pp., https://doi.org/19927, 1987.

Vitolo, C., Giuseppe, F. D., and Parrington, M.: Analysis and forecast of wildfires using ECMWF-Copernicus data and services, 13, 2491, 2019.

Wastl, C., Schunk, C., Lüpke, M., Cocca, G., Conedera, M., Valese, E., and Menzel, A.: Large-scale weather types, forest fire danger, and wildfire occurrence in the Alps, Agricultural and Forest Meteorology, 168, 15–25, https://doi.org/10.1016/j.agrformet.2012.08.011, 2013.

Woollings, T., Barriopedro, D., Methven, J., Son, S.-W., Martius, O., Harvey, B., Sillmann, J., Lupo, A. R., and Seneviratne, S.: Blocking and its Response to Climate Change, Curr Clim Change Rep, 4, 287–300, https://doi.org/10.1007/s40641-018-0108-z, 2018.

World Bank: World Bank Open Data, 2022.